# Population-based genome-wide association study of plasma complex lipid species

Elvire N. Landstra [1,7], Mohammed A. Imtiaz [1,7], Valentina Talevi[1], Fabian Eichelmann [2,3], Matthias B. Schulze [2,3,4], N. Ahmad Aziz [1,5,8] & Monique M. B. Breteler [1,6,8] ✉

The human lipidome comprises numerous complex lipids, dysregulation of which can contribute to the pathogenesis of a wide range of diseases. Despite the high heritability of parts of the lipidome, the genetic architecture of many circulating lipid species and their structure remains mostly unknown. Thus, we perform genome-wide association studies on 970 lipid species and 267 fatty acid composite measures using samples from the population-based Rhineland Study ($n = 6096$). We validate our findings using corresponding data from two other independent cohorts, including FinnGen ($n = 7266$) and EPIC-Potsdam ($n = 1188$). Out of 217 lead genomic loci, we find 136 to be novel, such as *FDFT1*. Using mendelian randomization and individual-level gene expression data, we identify 43 possible causal associations between candidate genes and corresponding lipid species, including *FDFT1* – diacylglycerol (16:0/18:0). Our findings provide new insights into the intricate genetic underpinnings of lipid metabolism, which may facilitate risk stratification and discovery of new therapeutic targets.

The metabolome—the collection of circulating metabolites—exhibits substantial inter-individual variability that may partly reflect differences in individual susceptibility to a variety of diseases, including cardio-metabolic and neurodegenerative disorders[1–3]. In particular, circulating levels of complex lipids, such as ceramides (CERs) and triacylglycerols (TAGs), are increasingly recognized as important correlates of age-related structural and functional changes in a variety of organs and tissues, including the brain, heart, kidney, and fat and muscle tissue[4–13]. Importantly, circulating complex lipid levels have also been associated with a range of common cardiovascular, metabolic and neurodegenerative diseases[14–16].

A large degree of the variation in the levels of circulating metabolites, including lipids, is thought to be genetically determined[17–19]. Indeed, previous genome-wide association studies (GWASs) have identified several genetic variants linked to circulating levels of certain triglycerides, phospholipids and sphingolipids[17,18,20,21]. These prior findings highlighted the importance of distinguishing between different lipid species rather than classes, as the genetic determinants of circulating lipid levels depended on their specific biochemical properties, such as the lipid species' carbon chain length and degree of saturation, rather than on their specific class[2,19]. However, with more than 26,000 lipids curated to date, much of their properties still remain to be elucidated,[22] including the genetic basis of the majority of complex lipid species, as well as their full fatty acid (FA) composition[17,18]. Indeed, the largest study to date employed a panel containing only 596 lipid species across 33 classes[23]. Therefore, there are a large number of lipid classes whose genetic basis has not been investigated at all. Furthermore, even for those lipids included in

[1]Population Health Sciences, German Centre for Neurodegenerative Diseases (DZNE), Bonn, Germany. [2]Department of Molecular Epidemiology, German Institute of Human Nutrition Potsdam-Rehbruecke, Nuthetal, Germany. [3]German Center for Diabetes Research (DZD), Neuherberg, Germany. [4]Institute of Nutritional Science, University of Potsdam, Nuthetal, Germany. [5]University of Bonn, University Hospital Bonn, Clinic for Parkinson's Disease, Sleep Disorders, and Movement Disorders, Bonn, Germany. [6]University of Bonn, University Hospital Bonn, Institute for Medical Biometry, Informatics and Epidemiology (IMBIE), Bonn, Germany. [7]These authors contributed equally: Elvire N. Landstra, Mohammed A. Imtiaz. [8]These authors jointly supervised this work: N. Ahmad Aziz, Monique M. B. Breteler. ✉e-mail: Monique.Breteler@dzne.de

previous GWAS studies, the genetic basis of their FA composition within and across classes was not systematically assessed.

As complex lipids have been critically implicated in the pathogenesis of many cardiometabolic and neurodegenerative diseases[14–16], the identification of their genetic determinants could improve our understanding of the molecular mechanisms underlying these disorders and yield potential new therapeutic targets. Therefore, taking advantage of recent technological innovations that enable accurate high-throughput profiling of a large number of lipid species that previously could not be measured at scale, we aimed to further disentangle the genetic architecture of complex lipid species and fatty acid composition, primarily focusing on those metabolites whose genetic determinants were so far unknown. We used data from the Rhineland Study, an ongoing population-based cohort study in Bonn, Germany, to perform a GWAS on circulating complex lipid levels and composition measured using the Metabolon Complex Lipids Platform. Importantly, we could validate our findings using corresponding data from two other independent population-based cohorts, namely the FinnGen and EPIC-Potsdam studies[24,25]. Subsequently, we contextualized our results through heritability estimates, phenome-wide association studies (PheWAS), and genetic correlation and mendelian randomization (MR) analyses. Finally, we also functionally validated several candidate genes using concomitantly available individual-level gene expression data. Our systematic approach uncovered a large number of novel genetic variants, metabolic quantitative trait loci (mQTLs), as well as candidate genes and pathways involved in complex lipid metabolism and FA composition.

## Results

### Sample characteristics

We included Rhineland Study participants who enrolled before the 26th of November 2021 ($n = 8318$), and who had complete lipidomic and genetic data available ($n = 6096$). The mean age was 55.9 years (range: 30–95), and 56.3% were women (Table 1). Comparing the included participants to those who were excluded because of missing data in any of the variables, we observed higher high-density lipoprotein cholesterol (HDL-C) ($p$-value < 0.001) and lower triglyceride levels ($p$-value < 0.001) (Table 1). An overview of the total lipid class concentrations can be found in Supplementary Table 1. To investigate connections between lipid species, we plotted correlations between lipids in a heatmap (Supplementary Fig. 1), most notably showing strong correlations among most TAGs.

### Replication cohorts

We used complete data from the FinnGenn cohort ($n = 7266$) and a subsample of the EPIC-Potsdam cohort ($n = 1188$) to replicate our findings. FinnGen and EPIC-Potsdam participants had an average age of 55.8 and 50.3 years and consisted of 64.7% and 60.3% women, respectively (Supplementary Table 2).

### GWAS of complex lipid species

We first ran a GWAS on lipid species adjusting for age, biological sex and the first 10 genetic principal components (model 1). Lipid species were quantified on the Metabolon Complex Lipids Platform, which identified 970 species across 14 lipid classes, where the number of species per class ranged between 12 and 518. Lipid species were characterized by a complete identification of the total number of carbons and double bonds and at least one FA tail (e.g., diacylglycerol DAG(16:1/20:0), ceramide (CER) 20:1, and triacylglycerol (TAG) 53:5(FA18:3)) (Supplementary Table 3).

In the Rhineland Study, we identified a total of 57 genomic loci associated with the levels of different lipid species after adjustment for sex, age and the first 10 genetic principal components. These genomic loci were identified by merging all metabolome-wide significant SNPs across all lipid species ($p < 2.27E-10$), followed by linkage

## Table 1 | Overview of the Rhineland Study sample characteristics comparing included versus excluded participants

| Characteristic | Included participants ($n = 6096$) | Excluded participants ($n = 2222$) | P-value[*] |
|---|---|---|---|
| Age (years) | 55.9 (13.7) | 55.8 (14.3) | 0.762 |
| Sex (women) | 3435 (56.3) | 1259 (56.7) | 0.806 |
| LDL-C (mg/dL) | 125.7 (35.6) | 124.3 (35.1) | 0.153 |
| HDL-C (mg/dL) | 63.0 (17.6) | 60.6 (17.3) | <0.001 |
| Triglycerides (mg/dL) | 109.2 (64.4) | 117.0 (84.1) | <0.001 |
| Cholesterol (mg/dL) | 199.2 (39.3) | 198.0 (39.0) | 0.315 |
| Use of lipid-lowering medication (yes) | 721 (12.0) | 253 (11.7) | 0.727 |

Data displayed represent the number of participants (percentages) for categorical variables and the mean (standard deviation) for continuous variables. Abbreviations: HDL-C = high-density lipoprotein cholesterol, LDL-C = low-density lipoprotein cholesterol.
*Adjusted for age and sex.

disequilibrium (LD) clumping to identify independent significant SNPs (using $r^2 \geq 0.6$ for defining independent significant SNPs) within 250 kb of each other, and 218 lead SNPs (using $r^2 \geq 0.1$ for the clumping of independent significant SNPs). For lead SNPs with no regional LD-supported signal, imputation quality (Rsq) and minor allele count (MAC) indicated high reliability, with a mean Rsq of 1.0 (range 0.86–1.0) and a mean MAC of 2841 (range 81–10,094), supporting the robustness of these associations (Supplementary Data 1). Out of the 57 identified loci (Supplementary Fig. 2a), 25 were novel across all lipid categories and classes (Fig. 1)—i.e., they neither matched previously reported loci in the mGWAS catalog, nor those reported by Cadby et al. and Ottsensmann et al[23,24]. Among these novel loci were *NIPAL1*, *ABCA7*, *GM2A*, *FDFT1*, *ITGA11*, which were most strongly associated with hexosylceramide (HCER) (FA22:1), lactosylceramide (LCER) (FA24:1), LCER(FA20:1)), DAG(16:0/18:2)) and phosphatidylethanolamine (PE) (18:0/18:3), respectively. The identified loci with the highest statistical significance were found on chromosome 11 (*FADS2*, *OR4C15*, *TRIM48*), followed by loci on chromosome 6 (*ELOVL2*, *GABBR1*). Although many genomic loci were unique to different lipid categories, some were also shared, particularly between neutral lipids and sphingolipids (Fig. 1). Specifically, we found lipidome-wide significant SNPs for 638 out of 970 lipid species. The lipid species within the sphingomyelins (SMs) (100% of 12 species), TAGs (78.2% of 518 species) and DAGs (75.9% of 58 species) classes had the highest number of genetic associations. On the other hand, the lipid species within the phosphatidylethanolamine ethers (PEOs) (30% out of 40 species) class had only a few genetic associations. The associated genomic loci and their independent significant SNPs identified in the GWAS on lipid species can be found in the Supplementary Data 1.

### GWAS of fatty acid composition

To evaluate the genetic determinants underlying biochemical features of lipids, we next considered the FA composite measures ($n = 267$). FA composite measures summarize the concentrations of all lipid species within a class of a certain length and degree of saturation (e.g., DAG(FA18:3)) (Supplementary Table 3). Additionally, we utilized information on the total FA composition. These total FA composites included all lipid species, regardless of class, of a specific length and degree of saturation (e.g., FA14:0).

In the GWAS on FA composite measures, we identified 51 genomic loci, of which 26 were novel, after adjusting for sex, age and the first 10 genetic principal components. Some of these were specific to a certain FA tail length or degree of saturation, while others were associated with lipids in general (Fig. 2). For example, the *FADS2*, *FADS3* and *ZNF259* (chromosome 11) and *APOE* (chromosome 19) loci were associated with FA composites of almost all lengths and degrees of

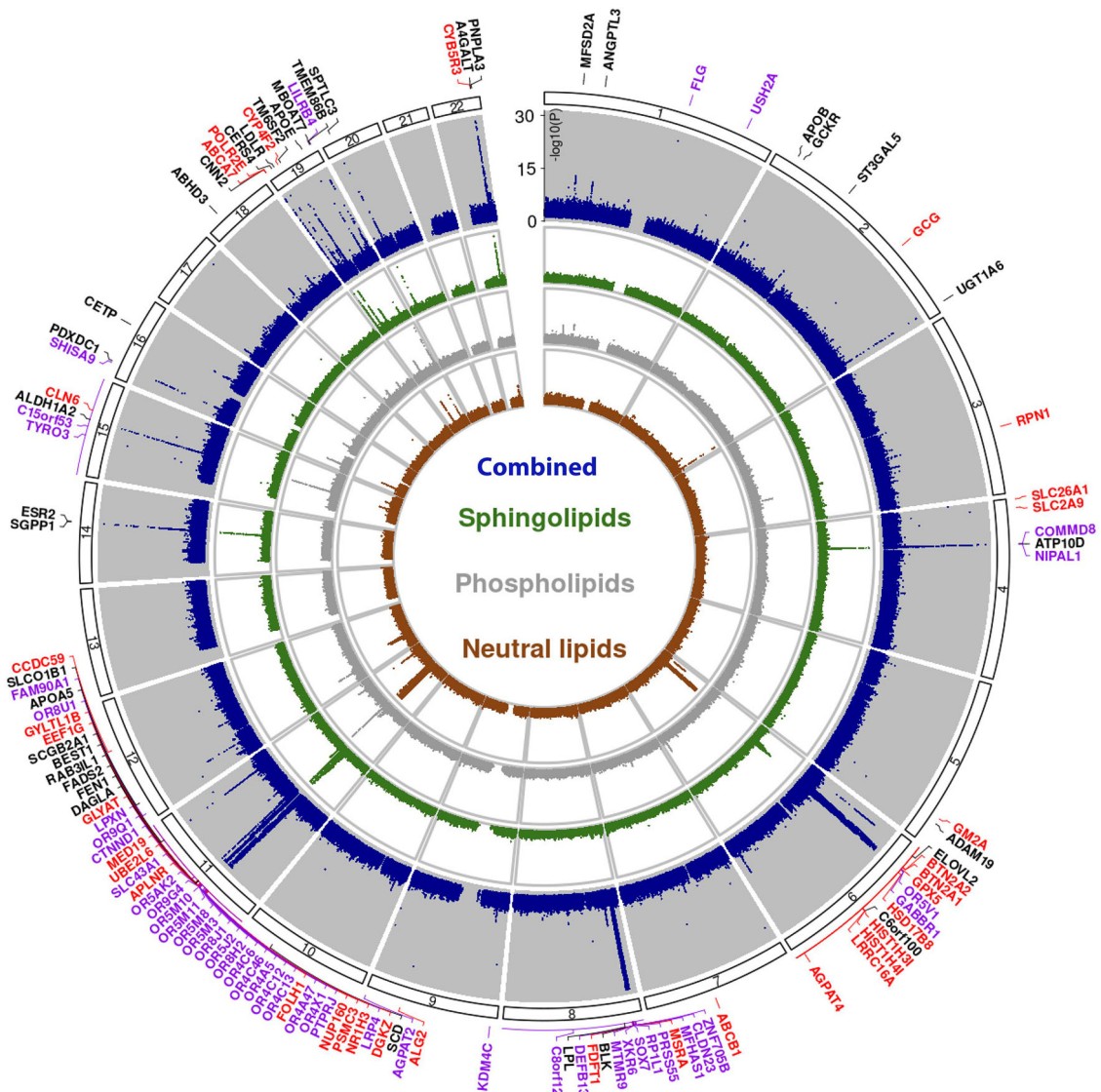

**Fig. 1 | Genomic loci associated with lipid species.** Circular Manhattan plot showing the effect loci (adjusted for age, sex and the first 10 genetic principal components), which are colored black when the lead SNP was identified previously, purple if the lead SNP is novel and linked to the genomic locus based on positional mapping and red if it is novel and linked to the locus based on in silico functional mapping. Lipid species are subdivided into their main categories (i.e., neutral lipids, phospholipids, sphingolipids, or combined), which are represented by the different circles. The p-value axis is truncated at $-\log_{10}(p) < 1e\text{-}30$ for visualization purposes. Analyses were performed on $n = 6096$ individuals, and each association estimate is derived from independent study participants.

saturation. In contrast, other loci were associated with lipids of a specific FA length, such as *PKD2L1* on chromosome 10 (16 carbons), *SGPP1* on chromosome 14 (14 and 22 carbons), *PDXDC1* on chromosome 16 (20 carbons), and *A4GALT* on chromosome 22 (22 and 24 carbons) (Supplementary Data 2). Similarly, some loci were unique to saturated FAs (e.g., *OR4C12*, chromosome 11), monounsaturated FAs (e.g., *C15ORF53*, chromosome 15; *CPS1*, chromosome 2; *PKD2L1*, chromosome 10), or both saturated and monounsaturated FAs (e.g., *ABCA7*, chromosome 19), whereas others were unique to polyunsaturated FAs, such as *SUGP1* (chromosome 19) (Supplementary Data 3). Across all different lengths, lipids carrying a FA tail of 24 (81.2% of 16 24-carbon carrying composites), 20 (74.2% of 65 20-carbon carrying composites) or 22 (63.6% of 54 22-carbon carrying composites) carbons had the highest percentage of significant genetic associations (Fig. 3).

The genomic loci and their independent significant SNPs identified in the GWAS on FA composite measures can be found in the Supplementary Data 4.

## Effect of adjustment for clinical lipid measurements, lipid-lowering medications and fasting status

All results described above were obtained after adjustment for sex, age, and the first 10 genetic principal components (model 1). To evaluate whether the hits were independent of clinical lipid measurements, and allow direct comparison of our results with those from a prior GWAS on lipids[23], we next adjusted our analyses for fasting status, HDL-C, total serum triglycerides, total cholesterol levels, and use of lipid-lowering medication (model 2). This would also enable us to assess whether the associations between genetic variants and lipid species simply reflect changes in lipid levels in general, or are specific to a certain species. After adjustment for clinical lipid measurements, use of lipid-lowering medication and fasting status, the effect estimates for the lipid species levels changed in a class-dependent manner (Fig. 4, Supplementary Data 5). Many SNPs were no longer associated with TAGs, DAGs and PEs, while the effect size estimates for the remaining associations became smaller. This is likely due to the genetic

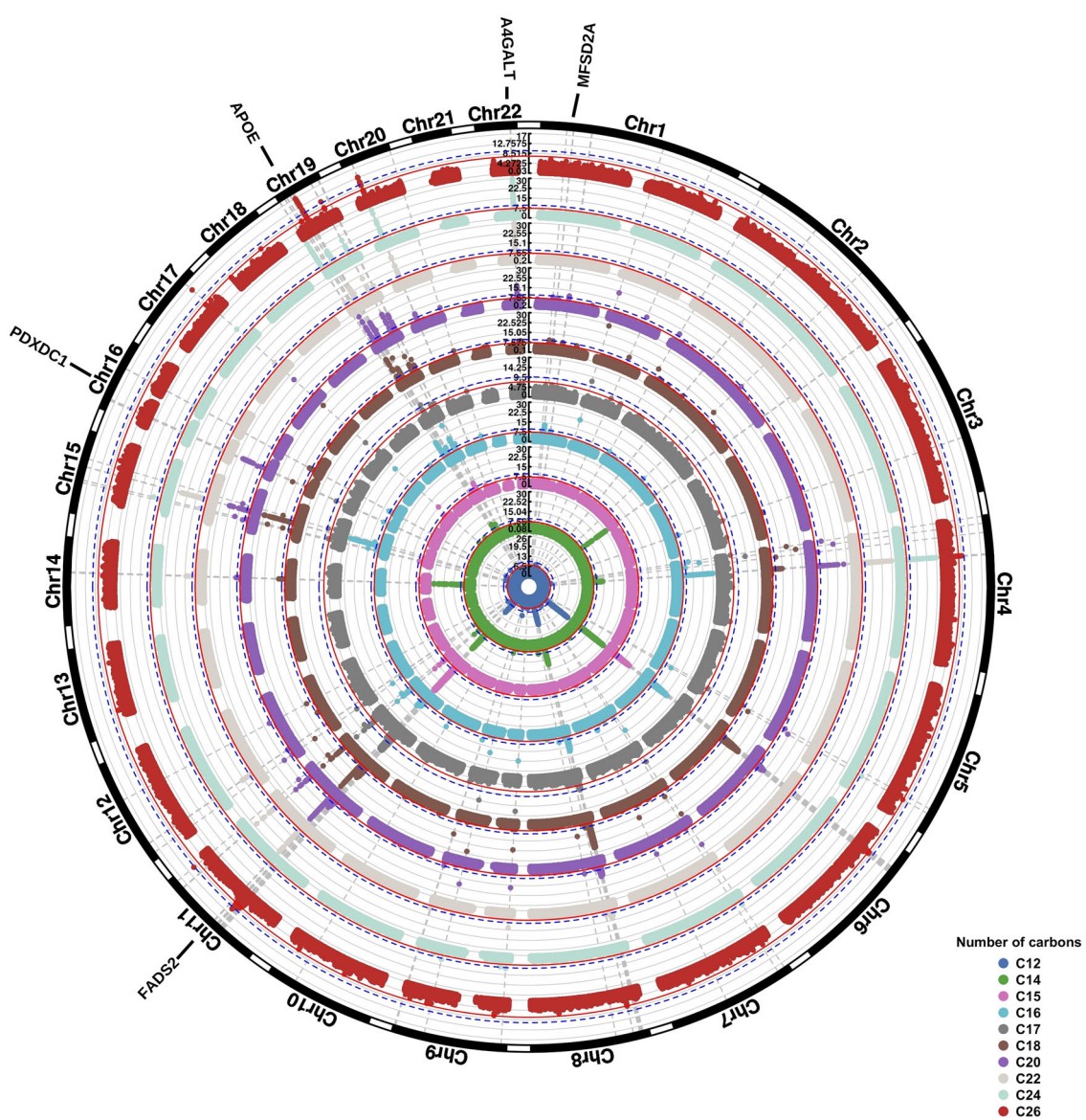

**Fig. 2 | Genetic associations of fatty acid composition based on the number of carbons.** Results of the GWAS on fatty acid composition (adjusted for age, sex and the first 10 genetic principal components) based on the number of carbons. The Manhattan plots are shown for the different fatty acid tail lengths separately, where the inner and outer circles represent the results for all lipids carrying fatty acid tails of 12 or 26 carbons, respectively. Analyses were performed on *N* = 6096 individuals, and each association estimate is derived from independent study participants.

correlation between these classes and the clinical lipids. As shown in Figs. 5–7, there is a substantial overlap between the genetics underlying lipid species belonging to these classes and the different clinical lipid measurements. After additional adjustment for clinical lipid measurements, use of lipid-lowering medication and fasting status, we identified a total of 61 lipid species-related genomic loci (Supplementary Fig. 2B), comprising 1248 metabolome-wide significant SNPs and 217 lead SNPs. Of these 61, 48 were already found in model 1. Newly identified genomic loci from the adjusted model (model 2) included *F3, CERS6, SLCO5A1, OR4C16, M6PR, ABHD2* and *TM4SF5*.

In contrast, for FA composite measures, the effect estimates across the different biochemical features (i.e., number of carbons and double bonds) remained similar to model 1 (Fig. 4). There were no changes with regard to the number of carbons or double bonds when comparing the results of model 1 with the adjusted model 2, which aligns well with the small number of significant genetic correlations between these biochemical characteristics and clinical lipid traits (Supplementary Fig. 3). All results for the adjusted model (model 2)

can be found in Supplementary Datas 6, 7, followed by model comparisons (Supplementary Data 8).

We additionally tested for evidence of effect modification by sex in the fully adjusted model (model 2). We found strong sex-dependent associations of the rs61763613 and rs11984568 SNPs with PEP(18:0/20:2) ($\beta = -2.54$, SE = 0.39, $p \approx 5.18\text{e}{-}11$) and TAG54:4(FA18:0) ($\beta = -0.238$, SE = 0.035, $p \approx 1.01\text{e}{-}11$), respectively (Supplementary Fig 4).

## SNP heritability
In order to evaluate how much of the variation in lipid species levels is explained by genetics, we estimated SNP heritability. After adjustment for age, sex, and the first 10 genetic principal components, lipid species showed a median SNP heritability estimate of 0.09 (interquartile range: 0.02 to 0.14) and exhibited substantial inter- and intra-class variability. Generally, we observed that HCERs (median: 0.15, interquartile range: 0.06 to 0.18) and SMs (median: 0.13, interquartile range: 0.12 to 0.14) had the highest estimated SNP heritability.

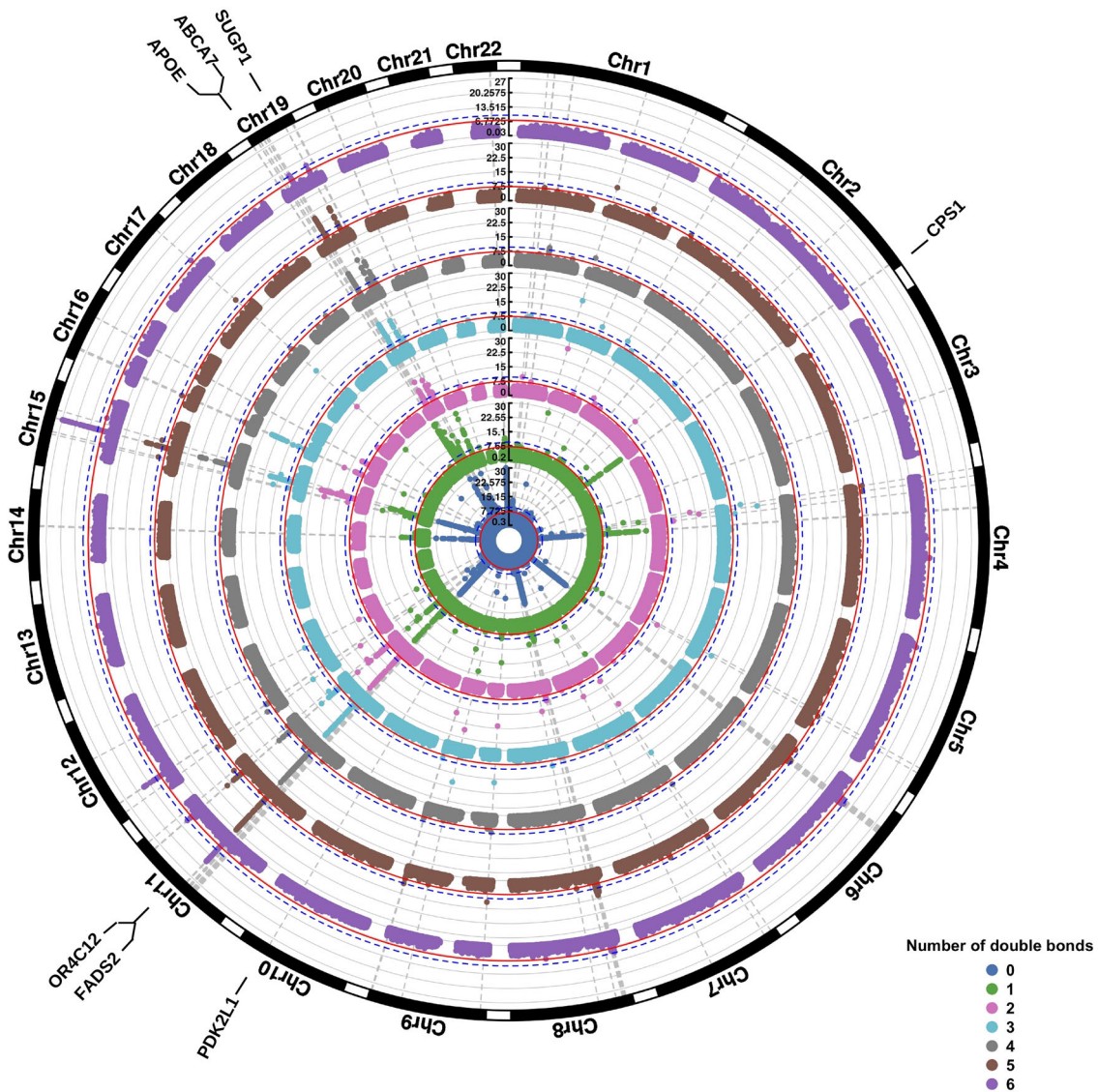

**Fig. 3 | Genetic associations of fatty acid composition based on the number of double bonds.** Results of the GWAS on fatty acid composition (adjusted for age, sex and the first 10 genetic principal components) based on the number of double bonds. The Manhattan plots are shown for the different fatty acid tail saturations separately, where the inner circle represents the results for all lipids carrying a saturated fatty acid tail and the outer circle for all lipids carrying fatty acid tails with 6 double bonds. All results were adjusted for age, sex and the first 10 genetic principal components. The $p$-value axis is truncated at $-\log_{10}(p) < 1e-30$ for visualization purposes. Analyses were performed on $n = 6096$ individuals, and each association estimate is derived from independent study participants.

However, the most highly heritable species did not belong to either of these classes, but to TAGs (TAG36:0(FA12:0), SNP heritability ($h^2$) = 1, SE = 0.05) and PEs (PE17:0/18:2, $h^2$ = 1, SE = 0.66). The least heritable classes were phosphatidylinositols (PIs) (median: 0.00, inter quartile range: 0.00 to 0.01) and lysophosphatidylethanolamines (LPEs) (median: 0.03, inter quartile range: 0.00 to 0.10), but specific species belonging to other classes also showed a low heritability (Supplementary Fig. 5A).

Next, we estimated the heritability of the FA composite measures separately based on their chain length and degree of saturation. After adjustment for sex, age, and the first 10 genetic principal components, we observed that within MAGs, the heritability was higher for saturated FAs (median: 0.63, range: 0.05 to 0.81), compared to unsaturated FAs, where those with six double bonds showed a median heritability of 0.18. On the other hand, cholesteryl esters (CEs) (median heritability: 0.10, range: 0.00 to 0.26) carrying a polyunsaturated FA had a higher heritability (median: 0.24, range: 0.09 to 0.24), compared to CEs carrying a saturated (median: 0.08, range: 0.03 to 0.16) or

monounsaturated FA (median: 0.05, range: 0.00 to 0.15). Similar to the CEs, TAGs carrying the most polyunsaturated FAs were most heritable (median: 0.16, range:0.13 to 0.18) (Supplementary Fig. 5B, Supplementary Data 9).

### Annotation of lead SNPs

To disentangle the mechanisms involved in lipid metabolism, we linked the 218 and 217 lead variants from models 1 and 2, respectively, to nearby genes (bottom-up approach–positional) and the ones that had previously been known to be related to metabolic pathways (top-down approach–Gene Ontology (GO), Kyoto Encyclopedia of Genes and Genomes (KEGG), mouse genome informatics (MGI), Orphanet, Reactome pathway databases). Specifically, after adjustment for sex, age, and the first 10 genetic principal components, 497 genes were identified through either the bottom-up or top-down approach. A total of 78 candidate genes overlapped between the two approaches, of which 31 were annotated to novel lead SNPs, and could therefore be classified as candidate genes in lipid metabolism. After additional

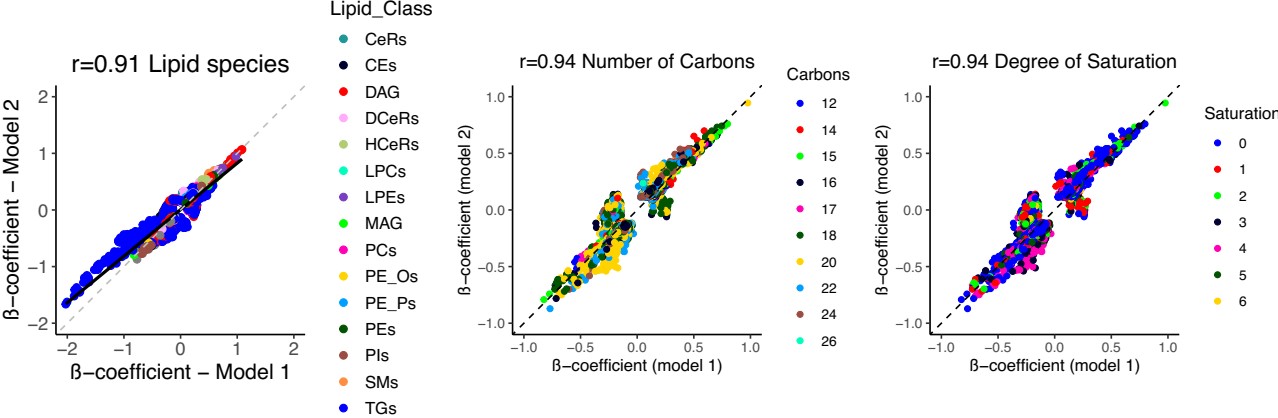

**Fig. 4 | Beta-beta plot.** Beta-beta plot depicting the ß coefficients of model 1 (x-axis) against those of model 2 (y-axis) for lipid species, including lipid composition categorized by the number of carbons and degree of saturation. The Pearson correlation coefficient (r) is indicated in the plot to quantify the strength and direction of the association between the effect sizes of the two models.

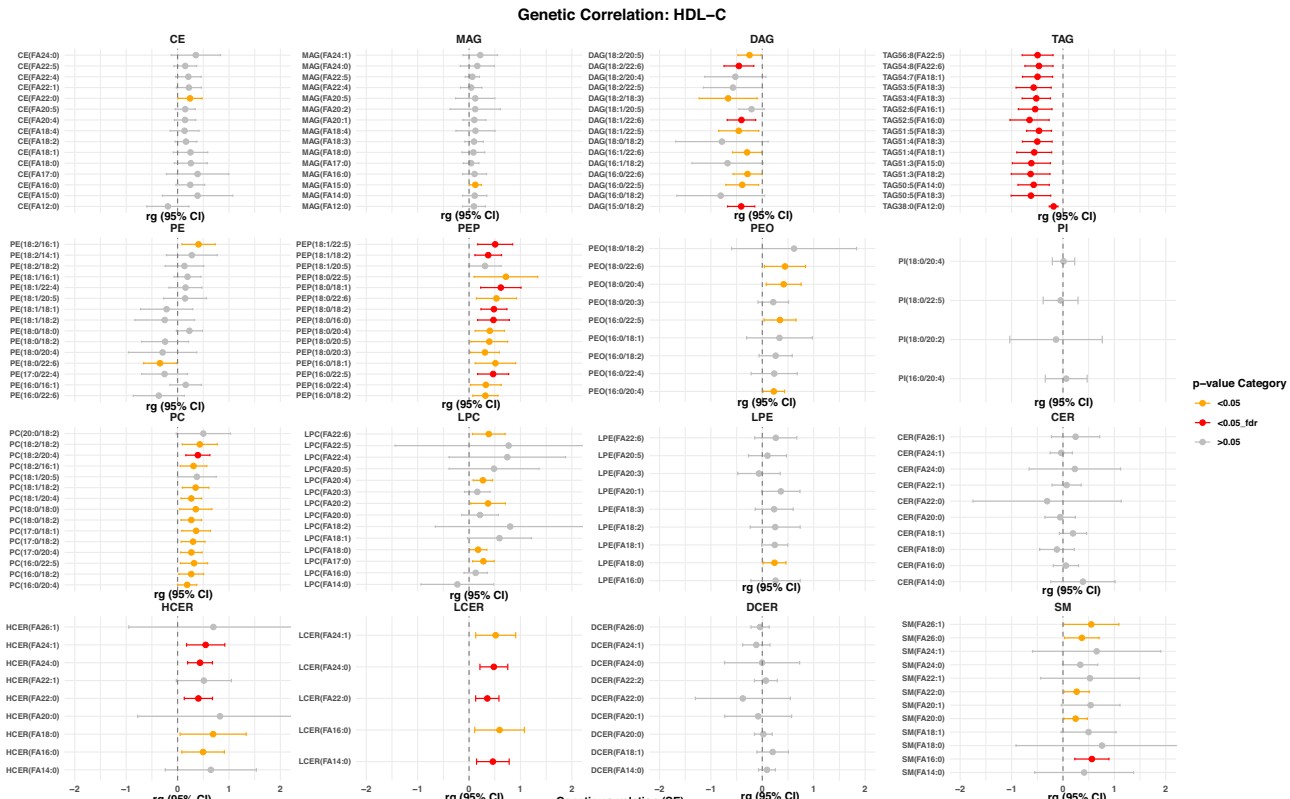

**Fig. 5 | Genetic correlations with HDL-C.** Genetic correlations of lipid species with HDL-C(model 1). The plot is restricted to the top 15 lipid species based on genetic correlation *p*-values. Dots represent the genetic correlation (rg)_estimates, while the whiskers indicate the corresponding 95% confidence intervals. Statistical significance is indicated by color, where red indicates a FDR-correct *p*-value < 0.05, orange a nominal *p*-value < 0.05 and gray a *p*-value > 0.05. Abbreviations: HDL-C high-density lipoprotein cholesterol.

adjustment for fasting status, clinical lipid measurements, and use of lipid-lowering medication, we identified a total of 523 genes, of which 84 were overlapping (Supplementary Data 10). Of these, 28 were annotated to a novel lead SNP. For FA composite measures, we found 64 and 66 candidate genes, of which 46 and 41, respectively, were novel (Supplementary Data 11). Overall, 61 of the identified candidate genes were shared among species and composite measures, which increased to 62 after adjustment for fasting status, clinical lipid measurements and use of lipid-lowering medication.

We investigated further whether the lead variants were also reported to be associated with gene expression levels in the Genotype-Tissue-Expression (GTEx) resource, and we used all the available 48 tissues from the GTEX v8 database, which could be classified as expression quantitative trait loci (eQTLs). Out of 218 lead variants associated with lipid species levels after adjusting for sex, age, and the first 10 genetic principal components (model 1), 53 were eQTLs, while out of 217 lead SNPs found after adjustment for clinical lipid measurements, use of lipid-lowering medication and fasting

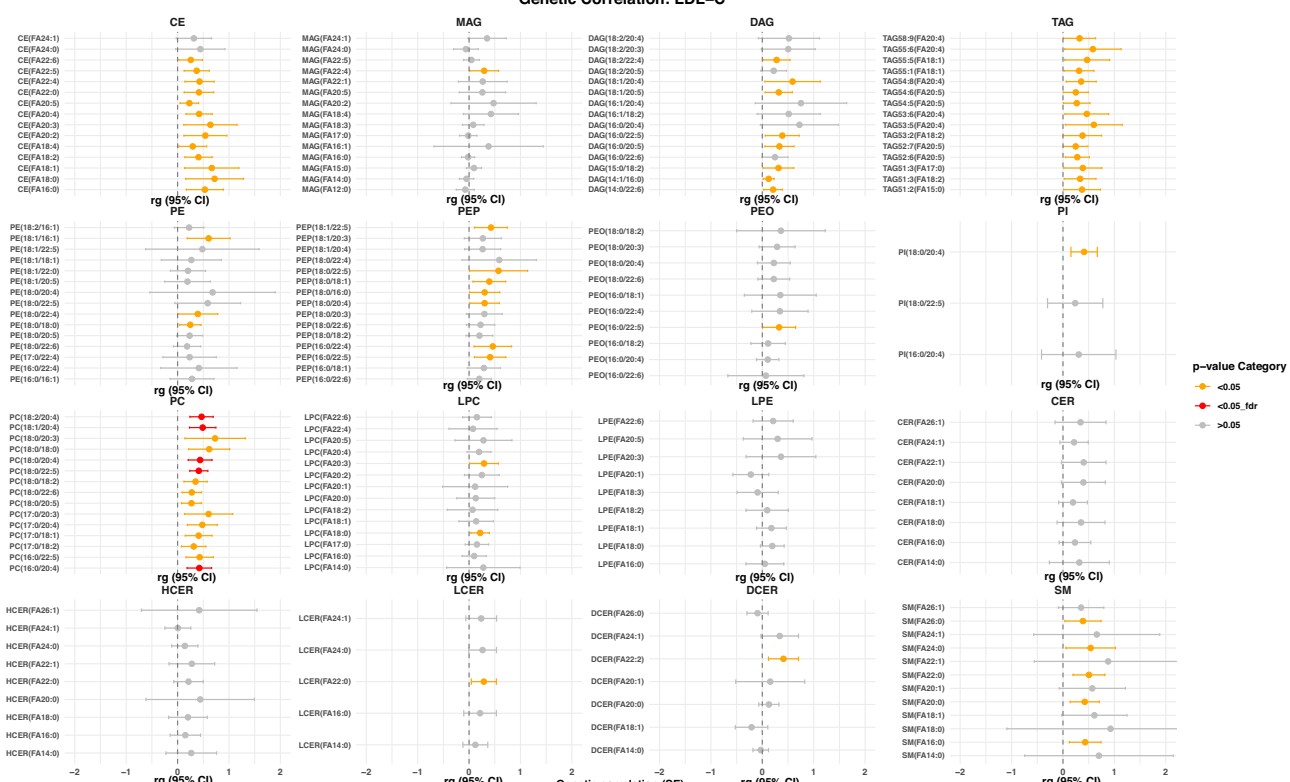

**Fig. 6 | Genetic correlations with LDL-C.** Genetic correlations of lipid species with LDL-C (model 1). The plot is restricted to the top 15 lipid species based on genetic correlation *p*-values. Dots represent the genetic correlation (rg)_estimates, while the whiskers indicate the corresponding 95% confidence intervals. Statistical significance is indicated by color, where red indicates a FDR-correct *p*-value < 0.05, orange a nominal *p*-value < 0.05 and gray a *p*-value > 0.05. LDL-C low-density lipoprotein cholesterol.

status (model 2), 89 were classified as eQTLs, which largely overlapped with those from the first model. Interestingly, our top hit (rs174547, *p*-value = 1e-226, beta estimate (β) = -0.626), which was most strongly associated with phosphatidylcholines (PC) (18:0/20:4), was found to be an eQTL for the *FADS2* gene. *FADS2* is a key player in lipid metabolism, regulating fatty acid desaturation, and was also found in several previous GWASs on lipids[1]. Our top novel hit (rs1589680, *p*-value = 1.9e-203, β = -0.637), which was most strongly associated with TAG(36:0(FA12:0)), was also an eQTL for *TRIM48*. For further validation of our findings, in addition to the eQTLs, we also assessed which of the lead SNPs associated with lipid species levels—which thus could be classified as metabolite quantitative trait loci (mQTLs)—were also reported previously. Sixty-five out of the 218 and 81 out of the 217 mQTLs identified in our analyses in models 1 and 2, respectively, were also detected in previous lipidomic GWAS studies[26].

For the composite measures, 46 eQTLs and 56 mQTLs were identified after adjustment for sex, age, and the first 10 genetic principal components, which changed to 59 eQTLs and 69 mQTLs upon further adjustment for traditional lipid measurements, use of lipid-lowering medication and fasting status. Similar to the results for the lipid species, the top hit (rs174546, *p*-value = 1.7e-198, β = -0.585) was most strongly associated with PC(FA20:4) and was an eQTL for *FADS2*. Some novel hits were also identified as eQTLs, including the rs3747193 SNP (*p*-value = 9.5e-63, β = -0. 492), which was mapped to the *A4GALT* gene and most strongly associated with LCER(FA22:0). The detailed overlap between the associations, which were previously reported in metabolomic-based GWAS studies and our lead SNPs, is provided in Supplementary Datas 10, 11.

## Validation in two independent cohorts

To validate our results, we first used data from a recently published GWAS on lipids in the FinnGen study (*n* = 7266)[24]. Out of 179 lipid species used in the GWAS from FinnGen, 78 lipid species were also quantified in the Rhineland Study (Supplementary Data 12). Of these, 61 lipid species were associated with at least one SNP at the metabolome-wide significant threshold in the Rhineland Study. For these lipid species, 677 SNP-lipid associations (involving 55 lipid species and 106 SNPs) were nominally significant in FinnGen, while only 76 associations were not. A total of 8652 associations (involving 1278 independent SNPs and 606 lipid species) could not be tested in FinnGen due to missing data on these SNP-lipid associations in FinnGen (Supplementary Fig. 3A, Supplementary Data 1). Vice versa, data on all 75 lipid species and 111 of FinnGen's lead SNPs were present in the Rhineland Study. Of the corresponding associations, 175 were replicated in the Rhineland Study, while only 33 were not (Supplementary Fig. 6A, Supplementary Data 12).

We also performed a replication analysis in another independent cohort that used the same lipid panel, namely the EPIC-Potsdam study (*n* = 1188)[25]. Out of 940 lipid species available in the EPIC-Potsdam study, 913 were also quantified in the Rhineland Study (Supplementary Data 13). Out of the 638 and 453 lipid species with at least one independent metabolome-wide significant association in the Rhineland Study in model 1 and model 2, respectively, EPIC-Potsdam had information on 638 and 432. Out of 8567 associations in model 1, 3479 were replicated across 563 lipid species and 447 SNPs (Supplementary Fig. 6B, C, Supplementary Data 1). In model 2, 2997 out of 6954 associations were replicated across 393 lipid species and 349 SNPs. (Supplementary Fig 6B, D, Supplementary Data 6). Across all lipid classes, we replicated an average of 67% of the SNP-lipid species associations,

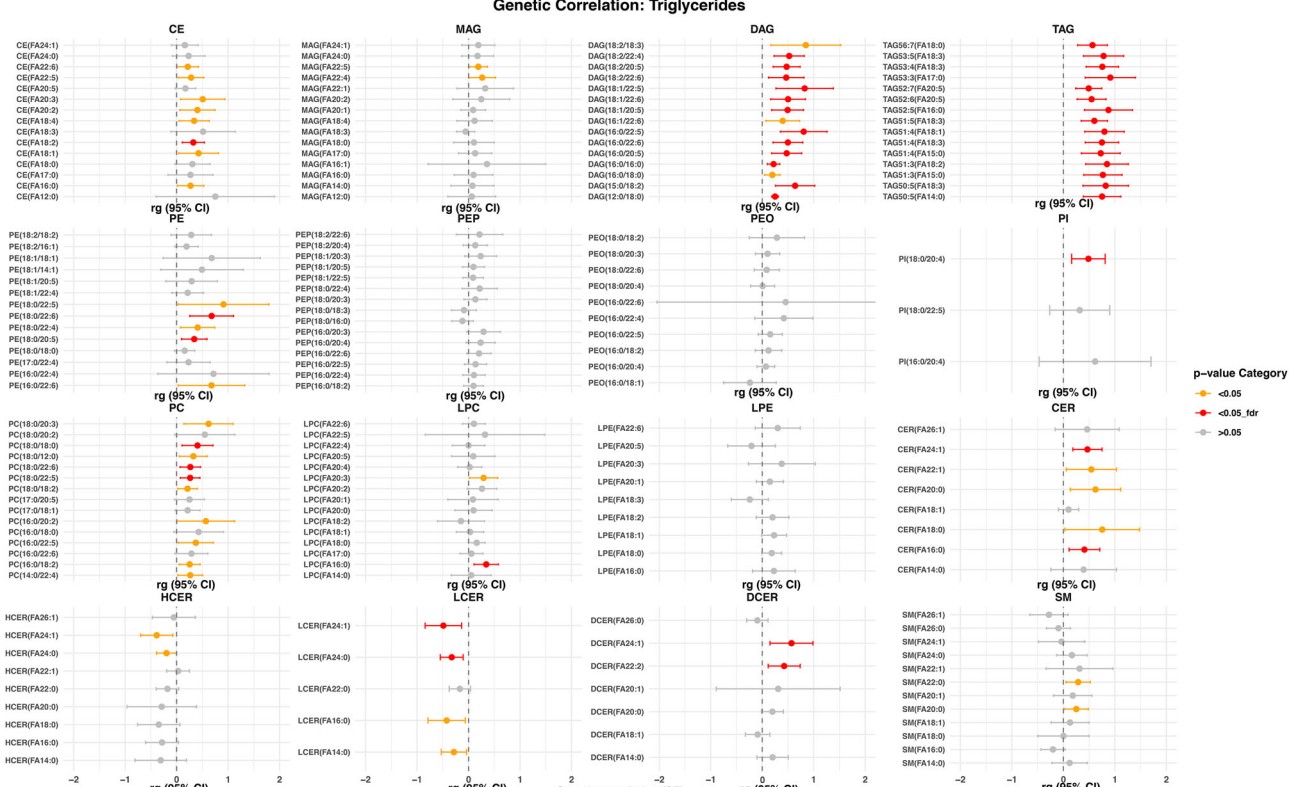

**Fig. 7 | Genetic correlations with total triglycerides.** Genetic correlations of lipid species with triglycerides (model 1). The plot is restricted to the top 15 lipid species based on genetic correlation *p*-values. Dots represent the genetic correlation (rg) _estimates, while the whiskers indicate the corresponding 95% confidence intervals. Statistical significance is indicated by color, where red indicates a FDR-correct *p*-value < 0.05, orange a nominal *p*-value < 0.05 and gray a *p*-value > 0.05.

where the number of SNP-lipid species associations that were replicated within a class ranged from 25% for DCERs to 95.5% for TAGs after adjustment for sex, age, and the first 10 genetic principal components, and from 60% for LCER to 100% for PEPs after additional adjustment for clinical lipid measurements, use of lipid lowering medication and fasting status (Supplementary Table 4). Overall, there was a high concordance between results, despite the smaller sample size of the EPIC-Potsdam Study (Supplementary Fig. 6C, D).

**Meta-analysis of results**
To integrate all findings, we meta-analysed the results from the Rhineland Study and EPIC-Potsdam. After adjustment for sex, age, and the first 10 genetic principal components (model 1), followed by adjustment for HDL-C, total serum triglycerides, total cholesterol levels, use of lipid-lowering medication and fasting status (model 2), we found a total of 64 genomic loci, 31 of which were novel in model 1, and 68 genomic loci, 40 of which are novel in model 2. Overall, the meta-analysis results were similar to the results from the Rhineland Study, except for the identification of 10 extra genomic loci in model 1 (*PAQR9, MLXIPL, SLCO5A1, TTC39B, SLC22A24, MOGAT2, M6PR, SOAT2, LIPG, GRAMD4*) and 9 extra loci (*SLC44A1, UGCG, C9orf91, C10orf82, SLC22A24, SEC14L5, LIPG, ATP8B1and GRAMD4*) in model 2 (Supplementary Data 14–16).

**Downstream analysis**
The following down-stream analyses are based on the results of the Rhineland Study GWAS conducted on lipid species after adjustment for traditional lipid measures, use of lipid-lowering medication and fasting status (model 2) in order to identify biological pathways underlying the effects of the lead SNPs and the corresponding genes that are independent of known clinical lipid traits.

**Phenome-wide association studies**
To further our understanding of how these candidate genes might affect health through lipids, we linked our results to disease phenotypes and clinical measures using PheWAS. We identified several important clusters of variant-trait associations for the 13 lead SNPs for which data were available (Fig. 8). Firstly, the lead SNP rs1047891 (*CPS1*) was associated with a variety of predominantly cardiometabolic traits and diseases, such as HDL-C levels, systolic blood pressure, glomerular filtration rate, diabetes and chronic kidney failure. Secondly, another important hub centered around rs603424 (*PKD2L*). This lead SNP was linked to phospholipid measurements, blood pressure, chronic kidney disease and cardiovascular diseases among others. Thirdly, rs738409 (*PNPLA3*) was previously linked to several traits, such as serum alanine aminotransferase levels, alcoholic liver cirrhosis, fatty liver disease and precursor cell lymphoblastic leukemia. Finally, although it was not identified as a major hub, the lead SNP rs4147929 (*ABCA7*), which was among the top hits related to fatty acid composition, was associated with Alzheimer's disease risk. Thus, these genes might affect disease phenotypes through changes in lipid metabolism. Additional information can be found in Supplementary Data 17.

**Colocalization and two-sample Mendelian Randomization analyses**
Guided by the results of our PheWASs, we performed additional colocalization analyses for lipid species-associated genetic variants mapping to the *ZPR1, APOE* and *SUGP1* loci and coronary artery disease[27], those mapping to *PNPLA3, SUGP1* and *ZPR1* loci and type 2 diabetes[28], those mapping to *APOE* and *ABCA7* loci and Alzheimer's disease[29], and those mapping to *FADS2, PKD2L1* and *CPS1* loci and chronic kidney disease[30]. Colocalization analyses revealed that the lipid species-associated genetic variants within the *ZPR1, APOE* and

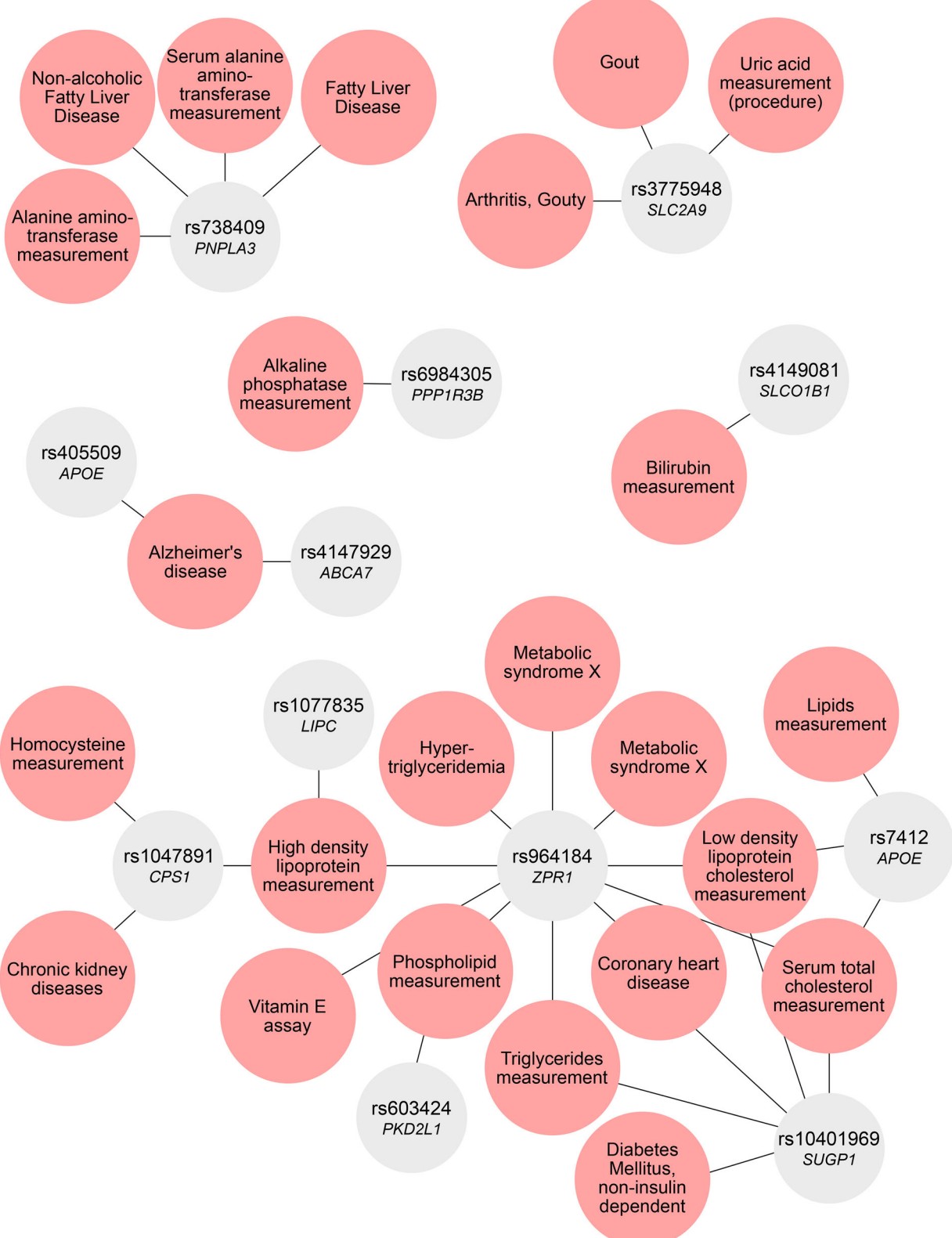

**Fig. 8 | Phenome-wide association studies' lookups.** The network graph depicts the results of phenome-wide association studies' lookups based on the lead single-nucleotide polymorphisms (SNPs) available in the DisGeNET database of gene- disease associations, showing phenotype-SNP associations, where the line indicates the connection between a disease and its mapped gene (variant-disease association score > 0.8).

*SUGP1* loci and coronary artery disease, those within the *SUGP1* locus and type 2 diabetes and those within the *CPS1* locus and chronic kidney disease, shared a single causal variant with posterior probabilities of the single shared causal variant (PP: H4) > 0.99 (Supplementary Data 18). In subsequent two-sample MR analyses, we also found evidence for potentially causal associations between 1–55 lipid species associated with the *PNPLA3*, *SUGP1* and *ZPR1* loci and coronary artery disease, with TAG42:0(FA16:0) levels exhibiting the strongest

association (IVW β = −0.32, SE = 0.03, FDR-adjusted $p$ = 4.5e-3), 2) five lipid species associated with the *APOE* and *ABCA7* loci and Alzheimer's disease, with the top hit being LCER FA24:1 (IVW β = -0.17, SE = 0.04, FDR-adjusted $p$ = 9.0e-3), 3) CE FA22:1 associated with the *CPS1* locus and chronic kidney disease (Wald ratio β = 0.34, SE = 0.61, FDR-adjusted $p$ = 3.2e-6), and 4) 75 lipid species associated with the *PNPLA3*, *SUGP1* and *ZPR1* loci and type 2 diabetes, with TAG 40:0 (FA14:0) exhibiting the strongest association (IVW β = -0.28, SE = 0.06, FDR-adjusted $p$ = 3.6e-5). All these associations were significant after FDR correction (Supplementary Data 19). Although reverse MR also hinted at several potentially causal associations in the reverse direction, these latter results were not reliable due to the presence of significant pleiotropic effects, except for Alzheimer's disease, which was causally associated with reduced levels of ceramide CER FA22:0 (IVW β = −0.13, SE = 0.04, FDR-adjusted $p$ = 4.5e-3) (Supplementary Data 19). We listed the SNPs that were used as instruments for two-sample MR in Supplementary Data 25.

### One-sample and two-sample Mendelian Randomization analysis of the association between gene expression levels and lipid species

Overall, through one-sample MR, we identified 43 potentially causal associations between the expression levels of 20 genes and 17 lipid species after adjustment for multiple testing using the FDR method. The strongest relation was between the expression levels of *FADS2* and PC(18:0/20:4) (β = −0.32, SE = 0.03, FDR-adjusted $p$-value = 2.5e-23). Among the novel candidate genes, we found evidence for a causal association of *FDFT1* (β = −0.9, FDR-adjusted $p$-value = 9.06e-6) and *BLK* (β = 0.99, FDR-adjusted $p$-value = 8.57e-12) expression with DAG(16:0/18:0). Furthermore, the MR analysis indicated a causal association between the expression levels of *MFHAS1* and DAG(12:0/18:0) (β = −1.14, FDR-adjusted $p$-value = 6e-5, F-statistic > 10) (Supplementary Data 20).

Out of the 523 genes identified through the top-down and bottom-up approaches, whole blood expression quantitative trait loci (eQTLs) were available for 239 genes in GTEx. Using these externally obtained eQTLs, we performed a two-sample MR analysis that largely confirmed the results of our one-sample MR analysis for 43 potentially causal associations between the expression levels of 20 and 17 lipid species (Supplementary Data 20). We listed the SNPs that were used as instruments for two-sample MR in Supplementary Data 21.

### Functional validation through gene expression analysis

We performed functional validation analyses using data from 2146 Rhineland Study participants for whom transcriptomic and lipidomic data were available. Out of the 523 genes identified through the top-down and bottom-up approaches, the expression levels of 239 were available in the Rhineland Study. The expression levels of 56 of these genes were significantly associated with 26 lipid species. The strongest association was observed between *FADS2* expression and PC(18:0/20:4) (β = −0.242, $p$-value < 0.001). The eQTL analysis showed 75 significant associations (FDR < 0.05) between lead SNPs and expression levels of primary candidate genes, with rs174564 as the top hit associated with *FADS2* expression (β = 1.0319, FDR < 0.001) (Supplementary Data 22). In addition, we performed mediation analyses using the expression levels of the genes ($n$ = 6) that were significantly associated with both lipid species concentrations and their corresponding lead SNPs. This analysis revealed that the expression levels of several candidate genes partially mediated the SNP-lipid species relationships, such as *FADS2*, which was the strongest mediator of the rs2727266-PEP(18:0/20:3) association (β = 0.06, $p$-value < 0.001, percentage of the effect mediated: 97%), and *ABCA7*, mediator of the rs4147909 – LCER(FA24:1) association (β = 0.03, $p$-value < 0.001, percentage of the effect mediated: 10.1%) (Supplementary Data 22). Overlapping associations between the mediation analysis and one-sample MR are displayed in Supplementary Data 23.

## Discussion

To elucidate the genetic architecture of circulating complex lipid levels and composition, we conducted a comprehensive GWAS encompassing a wide variety of lipid species and plasma FA composite measures. In the Rhineland Study ($n$ = 6096), we identified a total of 57 genomic loci before adjustment for fasting status, routine clinical lipid measurements (i.e., total serum triglycerides, HDL-C, and total cholesterol levels) and lipid-lowering medication, and 61 after additional adjustment for these measures. Importantly, more than one-third of the identified genomic loci were novel. Moreover, we discovered 84 candidate genes that were associated with lipid species independent of routine clinical lipid measurements and medication. By comparing our results to a previously published GWAS on a smaller set of lipids (FinnGen, $n$ = 7266), we found that we could replicate almost all of the previously reported associations (84.1%). Conversely, almost all of our metabolome-wide significant results, for which data were available in FinnGen, were replicated in this independent cohort (89.9%). We further demonstrated the robustness of our findings by running a second replication analysis in an independent cohort with the same lipid panel (EPIC-Potsdam, $n$ = 1188). Despite the smaller sample size, around 39.2% of our associations were replicated in the EPIC-Potsdam cohort as well. The meta-analysis GWAS of both cohorts resulted in the identification of ten additional loci in model 1 and nine additional genomic loci in model 2.

Importantly, we did not only assess lipid species levels, but also the overall FA composition of the lipidome, demonstrating that some genes were specific to a certain biochemical feature (i.e., FA tail length or degree of saturation), such as *SGPP1* and *A4GALT*, while others were associated with a variety of composite measures across the entire lipid spectrum, such as *FADS2* and *APOE*. We also found that the heritability of complex lipids showed high variability within classes and was mostly determined by a lipid's tail length as opposed to its class or degree of saturation. Although lipid species within the sphingolipids and neutral lipids showed substantial genetic overlap with clinical lipids, in line with previous studies[17], the genetic overlap with the number of carbons and double bonds was considerably lower. This lack of genetic overlap could account for our finding that the effect sizes of SNPs associated with levels of lipids belonging to these categories became non-significant upon adjustment for clinical lipids, while the effects of genetic variants associated with biochemical features of lipids were robust to further adjustment. Interestingly, our PheWAS analysis revealed links between the genetic modifiers of complex lipids and several diseases, including cardiovascular diseases and Alzheimer's disease, as well as other metabolic diseases such as type 2 diabetes. Our colocalization and Two-sample MR revealed that lactosylceramides are casually associated with Alzheimer's disease, cholesteryl esters are associated with chronic kidney disease, triacylglycerols are causally associated with cardiovascular disease and type 2 diabetes. Next, our MR analyses with our candidate genes yielded evidence for causal associations between the expression levels of a number of candidate genes identified in our in silico analysis and complex lipid levels. Moreover, our functional analysis demonstrated that up to 97% of the effects of several lead SNPs on lipid species were mediated through changes in gene expression.

Importantly, in contrast to most previous genetic association studies of complex lipids, we evaluated the genetic architecture of the whole plasma FA composition as well as the shared biochemical properties across different lipid species and classes, including their tail length and degree of saturation. By assessing FA composite measures, we could identify specific genes uniquely associated with certain biochemical properties of lipid composites. While genes like *FADS2*, *APOE* and *FLG* were associated with many complex lipids regardless of their precise biochemical properties, we found that others, such as *ABCA7*, were predominantly related to lipids with a specific degree of saturation, such as with LCERs that were either saturated or

monounsaturated FAs. Similarly, *PDXDC1* was related to lipids with specific tail lengths, such as with FAs carrying 20 carbons in LPCs, CEs and LPEs classes. In addition, we found that the heritability of FA composition was mainly determined by class or degree of saturation rather than by the lipid's tail length. This was mainly observed for MAGs, where heritability was higher for saturated FAs and lower for unsaturated FAs, whereas for CEs and TAGs, we observed the opposite. Collectively, our findings thus indicate that genetics play a crucial role in determining both the levels and biochemical attributes of complex lipids, extending previous findings that both aspects are pivotal for the effects of different lipid species[7,31–33].

Using one-sample MR analyses, we identified 20 genes that were potentially causally associated with 17 lipid species. These associations were largely confirmed in the additional two-sample MR analyses. Among these associations, we demonstrated that the expression levels of *FADS2*, *FADS1*, *CERS4*, *ABCA7* and *FDFT1* also partially mediate several associations between lead SNPs and lipid species' levels. Both *FADS2* and *FADS1* have a critical role in FA metabolism, which is further supported by our results[34,35]. *CERS4* was most consistently associated with total concentrations of sphingolipids, as well as individual sphingolipid species' levels and various FA lengths, which is biologically highly plausible given that *CERS4* encodes a ceramide synthase. Moreover, one previous study also specifically reported an association of genetic variants in *CERS4* with FA20:0, which we confirmed and expanded upon by showing an association with FA18:0 as well[36]. Interestingly, *ABCA7* was one of the most significant hits related to FA composite measures, especially those involving monounsaturated and saturated FAs, where its expression levels partially mediated the SNP-lipid associations. *ABCA7* encodes a lipid transporter and has previously been associated with an increased risk of Alzheimer's disease[37,38]. Moreover, recently, a potential causal role of LCERs in the relation between *ABCA7* and the risk of Alzheimer's disease was reported[39]. Our MR and mediation results extend these findings, indicating a pivotal role for *ABCA7* in the regulation of lipid metabolism, where its expression particularly controls monounsaturated and saturated lipids. Thus, we not only discovered a large number of mQTLs of complex lipids but also pinpointed genes whose expression levels are likely to causally mediate these associations.

Results of our PheWAS further highlight the important role of complex lipids as modifiers of a large number of different traits and diseases. For example, genetic variants in *CPS1* have previously been associated with a variety of different phenotypes, including HDL-C levels and diabetes[40]. The rare allele of the *CPS1* variant was previously related to a lower risk of coronary artery disease (CAD), specifically in women[41]. We found that *CPS1* was specific to monounsaturated cholesteryl esters, which are intimately linked to HDL-C as well as chronic kidney disease, diabetes and other diseases[13,42], and have previously been shown to have sex-specific effects[43]. Therefore, this link between *CPS1* and cholesteryl esters could explain previous findings on the risk of CAD. Another hub gene, *PKD2L*, regulates a crucial membrane protein and has previously been associated with cardiometabolic health as well as phospholipid measurements, among others. Although the mechanism linking this gene to lipids is still unclear, it has previously been linked to phospholipid measurements as well. We further extend these findings by showing that *PKD2L* mainly affects monounsaturated lipids and could be crucial for lipid metabolism, possibly through membrane interactions, as lipids are crucial parts of membranes as well. *PNPLA3*, on the other hand, was mainly associated with MAGs and TAGs in our analysis. Interestingly, this gene was previously reported in relation to alcoholic liver cirrhosis and fatty liver disease, which are characterized by an increase in triglycerides in the liver[44]. These results could pave the way to precision medicine and improved risk stratification. For example, measuring the identified lipids could not only aid in identifying individuals with a risk allele−without complex genetic analyses, but could possibly predict disease better than

the traditional lipid measurements in disease risk models. Furthermore, by identifying individuals with a genetic risk factor, stricter monitoring and regulation of these identified lipids could possibly aid in preventing disease. Therewith, it could improve precision medicine. In addition to the possible gene-lipid-disease links, some of our hits have also been linked to aspects of lipid metabolism before, such as the "biosynthesis of unsaturated FAs", which further strengthens our results. Thus, the PheWAS findings demonstrate that the genes related to our lead SNPs are at the center of different cardiometabolic traits and diseases and are involved in the regulation of specific aspects of lipid metabolism. It has to be noted, though, that certain diseases could also lead to changes in lipid profiles, rather than vice versa. However, we believe this to be less likely as the associations were identified in a mostly healthy population. Moreover, the results of our reverse MR analyses did not yield convincing evidence for this latter possibility. Therefore, the genes we identified might serve as potential targets for restoring disturbed lipid metabolism.

Our study has both strengths and limitations. First, although we present one of the largest and most comprehensive GWAS studies on complex lipid levels and composition to date, we did not perform experimental validation of our findings in model systems. In case of genetic heritability, although we accounted for genomic LD structure, it could be that residual population structure may have resulted in inflated heritability estimates for some lipid species. However, our systematic functional genomics approach identified several candidate genes, a large number of which we could functionally validate by using matched individual-level transcriptomics data. Moreover, we performed rigorous validation of our results, leveraging data from two other independent cohorts. Despite not being able to do this for all associations on account of missing data, we could validate most of our key findings. Another limitation concerns our MR analysis. Although we have identified several lipid species causally associated with specific diseases, the SNP instruments may be shared across neighboring lipid species, which limits the ability to determine true causal relationships between individual lipid species and diseases. Future studies using multivariable MR could help account for unmeasured pleiotropy and better identify true causal effects[45,46]. Additionally, although lead SNPs without LD partners exhibited relatively high Rsq and MAC, loci defined by a single variant are inherently more prone to false positive signals. Future studies with larger sample sizes or independent replication cohorts will therefore be important to validate these findings. The extensive, unique complex lipid data on a large number of individuals is a main strength of our study, as we could investigate the genetics of specific species, as well as the whole FA composition, which had been challenging so far. In particular, this was further strengthened by combining data from lipids measured on the same platform in both the Rhineland Study and EPIC-Potsdam cohort. However, some of our findings could not be directly compared to those of previous GWAS studies because of differences in lipidomics platforms used in each study. Nonetheless, despite the platform's large coverage, some lipids and lipid classes were not included, such as cardiolipins, phosphatidylserines and gangliosides. Furthermore, the complete biochemistry of certain lipids could not be confirmed, such as the total composition of the TAGs or the stereochemistry of FA tails.

In conclusion, we unraveled the genetic basis of a wide variety of circulating lipid species as well as FA composition. We discovered a large number of novel mQTLs for complex lipids, identified candidate genes that could be crucial in regulating specific aspects of lipid metabolism, and provided evidence for their clinical relevance by linking them to a diverse array of cardiometabolic traits and diseases.

## Methods

Approval to undertake the Rhineland Study was obtained from the ethics committee of the University of Bonn, Medical Faculty. The study protocol of EPIC-Potsdam was approved by the ethics committee of

the Medical Society of the State of Brandenburg, Germany, and all participants provided a statement of written informed consent before enrollment. Both studies were carried out in accordance with the recommendations of the International Council for Harmonization Good Clinical Practice standards. Written informed consent was obtained from all participants in accordance with the Declaration of Helsinki.

## Study population

We used data from the Rhineland Study, an ongoing prospective cohort study in Bonn, Germany. People aged 30 or above who lived in either of two geographically defined areas in Bonn were invited to participate. Participants are invited in random order, based on postal code. The only exclusion criterium was having insufficient command of the German language to provide informed consent. Participants underwent deep phenotyping to obtain genetic, imaging, socio-demographic, clinical as well as metabolic information. Participants who enrolled before the 26th of November 2021 ($n = 8318$) were included when they had complete lipidomic and genetic data ($n = 6096$).

In addition, data from the independent EPIC-Potsdam study were used for a replication analysis. EPIC-Potsdam is a prospective cohort study in Potsdam, Germany, that started in 1994 and includes ~27,500 participants between the ages of 35 and 64 at baseline[25]. Data was collected on diet, anthropometrics, genetic and lipidomic measures, among others. In the current analysis, all participants from a random subsample ($n = 1188$, ~5% of total participants) with complete genetic, lipidomic, and clinical lipid data available were included.

## Lipidomics

Absolute (nmol/ml) plasma lipid concentrations were measured using the Metabolon Complex Lipids platform. Almost all blood samples were collected after an overnight fast (fasting: $n = 6045$ fasting, non-fasting: $n = 51$). Samples were stored at -80 degrees Celsius until further processing and analysis.

An automated Butanol–Methanol (BUME) method was used for lipid extraction[47]. First, extracts were dried and reconstituted in ammonium acetate dichloromethane:methanol, which was then transferred for analysis in a Shimadzu Liquid Chromatography with nano PEEK tubing and the Sciex Selexlon-5500 QTRAP mass spectrometer. Lipid extracts were analyzed using flow injection analysis-tandem mass spectrometry (FIA-MS/MS) with differential mobility spectrometry (DMS) separation and multiple reaction monitoring (MRM). For lipids that have unique precursor/fragment ion pairs (cholesteryl esters, ceramides, diacylglycerols, dihydroceramide, hexosylceramides, lactosylceramides, monoacylglycerols, and triacylglycerols), DMS separation was not required.

To extract various different lipids, both positive and negative mode electrospray were used. By multiplying the concentration of the internal standard with the ratio between the signal intensity of the target compound and the isotopically labeled internal standard ($n = 54$), the absolute concentration of 1023 lipid species was quantified, grouped into neutral, phospho- and sphingolipids based on their fundamental biochemical structural properties[48]. Samples were analyzed in sets of up to 40 samples, of which at least four were technical replicates and three process blanks. Data post-processing included background subtraction and normalization to correct for run-day effects.

Besides individual species, FA composite measures ($n = 271$) were also calculated. FA composite measures summarize the concentrations of all lipid species that have at least one FA tail of a specific length and saturation. Lipid species were characterized by a complete identification of all FA tails for all lipid classes except TAGs (e.g., DAG(16:1/20:0, CER20:1) (Supplementary Table 3). For the TAGs, which have 3 FA tails, lipids were considered species when information on the total number of carbons and double bonds, as well as detailed information on one FA tail, was available (e.g., TAG53:5(FA18:3)). A more detailed overview with examples of the lipid nomenclature used in this paper can be found in Supplementary Table 3. Only lipids present in at least 1% of the population were evaluated in the current analysis ($n = 970$). Missingness per lipid species is included in Supplementary Data 24. Pearson's correlation coefficients were calculated to evaluate the pairwise associations between different lipids.

## Genomics

Blood samples were genotyped using the Illumina Omni-2.5 exome array, which contains 2,612,357 single-nucleotide polymorphisms (SNPs). Subsequently, genotype data were processed using GenomeStudio (version 2.0.5) and quality controlled with PLINK (version 1.9). SNPs were excluded if not meeting the Hardy-Weinberg disequilibrium criterium ($p$-value < 1E-5), having a minor allele frequency <0.01, or showing a poor genotyping rate (< 99%). Additionally, participants were excluded because of a poor call rate of less than 95% ($n = 41$), abnormal heterozygosity ($n = 69$), cryptic relatedness ($n = 261$), or a sex mismatch ($n = 28$). To account for variation in population structure, which may otherwise cause systematic differences in allele frequencies[49], we used EIGENSTRAT (version 16000)[49]. EIGENSTRAT uses principal components to detect and correct for population structure, which resulted in the exclusion of an additional 164 participants from non-Central European descent. Finally, we imputed missing SNPs with IMPUTE (version 2)[50] based on the 1000 Genomes (phase 3) reference panel[51]. We ensured a high imputation quality by only including SNPs with an info score metric > 0.3, which indicates reliable imputation[50].

## Transcriptomics

We functionally validated our results using gene expression data of the first 3000 consecutively enrolled participants of the Rhineland Study. Samples used for RNA sequencing were stored in PAXgene Blood RNA tubes (PreAnalytix/Qiagen), and were thawed and incubated at room temperature to increase RNA yields. Total RNA was isolated according to the manufacturer's instructions using PAXgene Blood miRNA Kit following their automated purification protocol (PreAnalytix/Qiagen). RNA integrity and quantity were evaluated using the TapeStation RNA assay on a TapeStation4200 instrument (Agilent). The Tapestation RNA assay (Tapestation4200 instrument from Agilent). After using 750 ng of total RNA to generate next generation-sequencing libraries for total RNA sequencing (TruSeq stranded total RNA kit, Illumina), a Ribo-Zero Globin reduction was performed. Libraries were quantified using Qubit HS dsDNA assay (Invitrogen) and clustered at 250 pM concentrations on a NovaSeq6000 instrument. Quality control of the sequencing reads was performed with FastQC (v0.11.9). Following the filtering of low-quality score reads with Trimmomatic (v.0.39), sequencing reads were aligned to the human reference genome GRCh38.p13 using STAR (v2.7.1). The count matrix was generated with STAR –quantMode GeneCounts using the human gene annotation version GRCh38.101. Genes with overall mean expression > 15 reads and expressed in at least 5% of the participants were retained for further analysis. Raw counts were normalized and transformed using the varianceStabilizingTransformation function from DESeq2 (v1.30.1).

## GWAS

We performed a GWAS on the absolute concentrations of lipid species and FA composite measures using Rvtests[52]. In the first model, we regressed out the effects of age, self-reported biological sex and the first 10 genetic principal components to account for population structure (model 1). In the second model, we additionally adjusted for fasting status, as well as the levels of total cholesterol, triglycerides, HDL-C, and the use of lipid-lowering medication (model 2). Adjusting for clinical lipid measures enabled us to assess whether genomic loci

were associated with overall lipid levels or were more specifically related to a certain lipid species. Residuals obtained in either model were subsequently normalized using a rank-based inverse normal transformation before performing the GWAS. In the second and third models, only the participants with complete data on the additional covariates were included. To test for evidence of effect modification by sex, we included a sex-interaction term in model 3. We set the genome-wide significance level at $p < 5E-8$. Because of the large number of outcome variables (i.e., adjusted metabolite levels), we corrected for multiple testing using the method of Li et al.[53], which accounts for the correlation between the lipid levels by estimating the effective number of independent tests. Accordingly, we estimated the effective number of independent tests to be 222, resulting in a metabolome-wide significance level of $p < 2.27E-10$ ($\approx 5E-8/222$).

For further functional mapping of genomic loci that influence lipid metabolism, we aggregated all GWAS results in a unique dataset, in which we included the lowest $p$-value across all lipids for each SNP as described previously[23]. Next, the aggregated data were processed with the Functional Mapping and Annotation (FUMA) tool[54] to identify the lead and independent SNPs associated with lipid species and FA composite measures. We used the 1000 Genomes dataset (phase 3) as a reference panel to account for the linkage disequilibrium (LD) structure of the metabolome-wide significant genetic variants and identification of independent metabolome-wide significant SNPs (squared allelic correlation ($r^2$) $\geq 0.6$). Independent lead metabolome-wide significant SNPs were defined as those genetic variants with $r^2 \geq 0.1$. Each genomic risk locus was defined by tagging independent significant SNPs in close physical proximity (i.e., within 250 kb from either side of each LD block). The genomic risk loci thus contain multiple independent significant and lead SNPs.

## Annotation of lead SNPs

We systematically identified putative causal genes involved in lipid metabolism using both the top-down and bottom-up approaches within the ProGeM framework[55]. The top-down approach uses the curated database of known metabolic-related genes to identify biologically relevant genes that are present within 500 kb of the lead SNP. The bottom-up approach identifies genes based on genomic distance, where the three closest protein-coding genes near the lead SNP are selected. We then used the impact factor score, as calculated by the Variant Effect Predictor (VEP) method[56], to classify the lead SNP's function as either missense, start loss or stop gain. Primary candidate genes for lipid metabolism were defined as those genes that were identified through both the top-down and the bottom-up approaches. We also used mGWAS-Explorer[26] and PhenoScanner[57] to assess whether our lead SNPs had been identified previously in other lipidomic GWAS studies.

## Genetic architecture of complex lipids

To investigate the genetic heritability of the lipid species and FA composition, we calculated the SNP-based heritability using Genome-wide Complex Trait Analysis (GCTA)[58]. For this, we took the rank-based inverse-transformed residuals of lipid species and FA composites as phenotypes obtained from model 1 and model 2 in the Rhineland Study. In addition, we used LD Score Regression[59] to calculate the genetic correlation between different complex lipids. For the genetic correlation analysis, we used the summary statistics from the largest GWAS studies on three traditional lipid traits[27,60–62] (including HDL-C, LDL-C and triglycerides). A false discovery rate (FDR) < 0.05 was considered statistically significant.

## Validation of GWAS results in independent cohorts and meta-analysis

First, we performed validation of our GWAS results using summary statistics derived from a recently published large-scale GWAS study on complex lipids, including 179 lipid species, in the FinnGen cohort[24]. We compared whether our metabolome-wide significant SNPs replicated at a nominally significant level ($p < 0.05$) in the FinnGen GWAS summary statistics. In addition, we evaluated whether the lead SNP associations reported in the FinnGen cohort replicated in our study. Second, we validated our results by running a replication analysis in the EPIC-Potsdam study, which uses the same lipidomic platform. Similar to the approach above, we checked whether our metabolome-wide significant SNPs were nominally significant ($p < 0.05$) in a separately-run GWAS on complex lipids in the EPIC-Potsdam cohort. Lastly, to integrate all results, we performed a meta-analysis for overlapping data of the Rhineland Study and the EPIC-Potsdam cohort.

## Phenome-wide association studies

To further assess the clinical relevance of our findings, we also performed PheWAS analyses. We used the DisGeNET database to assess the relation between the lead SNPs associated with lipid species, after adjustment for traditional lipid measures, and the aforementioned phenotypes. Using the disgenet2r package, we obtained Variant-Disease Association (VDA) scores, which translate the certainty of the results into a score ranging from 0 to 1[63]. We only report the phenotype-SNP associations that had a VDA score > 0.7.

## Colocalization and two-sample Mendelian Randomization Analyses

We performed colocalization analyses to determine whether the genetic variants associated with lipid species overlapped with those related to disease traits identified in our PheWAS analyses. For each lipid trait, we considered variants within ±500 kb of the lead SNPs and merged them with summary statistics from publicly available GWAS data on coronary artery disease, Alzheimer's disease, chronic kidney disease and type 2 diabetes. Colocalization was conducted using the coloc.abf function from the coloc R package[64], which estimates posterior probabilities for five hypotheses: (H0) neither trait is associated with the region; (H1/H2) only one trait is associated; (H3) both traits are associated, but with distinct causal variants; and (H4) both traits are associated and share a single causal variant. The prior probability for a shared causal variant (H4) was set at $1.0E10^{-6}$. Posterior probabilities were summarized across all lipid traits to identify top colocalizing regions, with particular attention to traits showing strong evidence of colocalization (e.g., PP.H4 > 0.9). We further conducted two-sample Mendelian Randomization (MR) analyses using the TwoSampleMR package[65]. Lipid species were used as exposures, while the disease(-related) traits identified in our PheWAS analyses were used as the outcomes. For each exposure and outcome, only genome-wide significant SNPs were utilized. Harmonization of exposure and outcome datasets was performed to align effect alleles. MR estimates were obtained using inverse variance weighted regression.

## One-sample mendelian randomization

We employed a one-sample MR approach to test for evidence that the associations between genes identified through our top-down and bottom-up approach and lipid species (adjusted for clinical lipid measures) were causal, using the OneSampleMR package (version 0.1.5)[66]. To this end, gene expression levels were first adjusted for age, sex, batch effects, red and white blood cell counts, and the fractions of basophils, eosinophils, lymphocytes, monocytes, and neutrophils, and residuals were normalized using a rank-based inverse normal transformation. Subsequently, for each gene of interest (exposure), we run a GWAS to identify genome-wide significant SNPs associated with its expression levels that could serve as genetic instruments. Lipid species were treated as the outcome. We clumped SNPs in LD ($r^2 < 0.01$ within a 10 Mb window). Prior to analysis, lipid species levels were adjusted for age, sex, total cholesterol, triglycerides, HDL-C, lipid-lowering medication use, fasting status, and the first 10 genetic PCs. The residuals from this model were also normalized using a rank-based inverse

normal transformation. The MR analysis was then performed using two-stage least squares (2SLS) instrumental variable regression. In this framework, the outcome (lipid species) is regressed on covariates and gene expression levels, while gene expression is instrumented by the selected SNPs. This approach allowed us to estimate the causal effect of gene expression on lipid species while accounting for potential confounding. We set the statistical significance level at FDR < 0.05 for identifying significant causal associations. Our one-sample MR analysis relies on the three-core instrumental-variable assumptions: 1) relevance—the selected SNPs are associated with gene expression levels, 2) independence—the selected SNPs are not associated with confounders of the gene expression-lipid species relationship; and 3) exclusion restriction—the selected SNPs influence lipid species only through gene expression and not through other pathways. These assumptions justify the use of SNPs as valid instruments in the two-stage least squares framework.

### Two-sample Mendelian randomization

In complementary analyses, we also employed a two-sample MR approach to assess the robustness of the potentially causal associations between the genes and lipid species identified in our one-sample MR analyses, using the TwoSample MR package, because, compared to one-sample MR, two-sample MR is less prone to overfitting. We downloaded whole blood eQTL data from the GTEx resource[67], and used variants associated with gene expression levels as genetic instruments in the MR analyses. Specifically, eQTLs of candidate genes were used as the exposure and variants associated with lipid species levels as the outcome. We clumped SNPs in LD ($r^2 < 0.01$ within a 10 Mb window). The method selection for Two-sample MR was based on the number of SNP instruments available, including Wald Ratio (1 SNP instrument), inverse variance weighted method (> =2 SNP instruments), and MR-Egger method (> 3 SNP instruments). To assess the robustness of our MR results, we performed several sensitivity analyses, such as calculating F-statistics to assess the risk of weak-instrumental bias and MREgger intercept to identify horizontal pleiotropy. Our two-sample MR analyses rely on the three-core instrumental-variable assumptions: relevance—the eQTL variants used as instruments are robustly associated with gene expression, as supported by F-statistics; independence—the instruments are not associated with confounders of the gene expression–lipid species relationship; and exclusion restriction—the instruments influence lipid species only through gene expression. The complementary two-sample MR analyses additionally assume that SNP–exposure effects estimated in GTEx whole-blood eQTL data are transportable to the lipid species GWAS population, that there is no sample overlap between exposure and outcome, and that linkage disequilibrium between instruments is adequately controlled ($r^2 < 0.01$). Sensitivity analyses further rely on the assumptions underlying weak-instrument diagnostics and the MR-Egger method, which requires that SNP instruments' effects on the exposure are independent of any direct pleiotropic effects.

### Functional validation through gene expression analysis

For functional validation of the primary candidate genes, we first investigated the association of lipid species levels (dependent variable) with the residuals of candidate genes' expression levels after adjustment for age, sex, batch, red and white blood cell counts, as well as the fraction of basophils, eosinophils, lymphocytes, monocytes, and neutrophils (independent variables) using linear regression. The analysis was conducted on data from 2146 participants of the Rhineland Study for whom gene expression and lipid species data were available. For this validation step, we set the statistical significance threshold at $p < 0.05$. Secondly, we conducted an eQTL analysis for the lead SNPs with the residuals of candidate genes' expression levels after further adjustment for the first 10 genetic PCs, including participants with complete gene expression and genotype data ($n = 2008$), setting the statistical significance level at FDR < 0.05. Lastly, for combinations where the gene-lipid and SNP-gene associations were significant, we conducted mediation analyses using the R package lavaan (v.0.6-11) to assess whether gene expression levels (mediator) mediated the relationship between the lead SNPs (exposure) and lipid species levels (outcome). Specifically, we performed structural equation modeling (SEM), using a standard three-variable mediation model: path a represented the effect of the SNP on the gene expression, path b the effect of the gene expression on lipid levels, and path c' the direct effect of the SNP on lipid levels. Indirect effects were calculated as a*b, the direct effect as c', and the total effects as a*b + c. The model was specified in lavaan using the sem.fit function, using bootstrapping (with 1000 resamples) for calculating confidence intervals. For the mediation analyses, we used the residuals of gene expression adjusted for age, sex, batch, and blood cell counts, and residuals of lipid species adjusted for age, sex, the first 10 principal components, total cholesterol, triglycerides, HDL-C, lipid-lowering medication use, and fasting status.

### Reporting summary

Further information on research design is available in the Nature Portfolio Reporting Summary linked to this article.

## Data availability

The data from the Rhineland Study and EPIC-Potsdam used in this manuscript are not publicly available due to data protection and privacy regulations. For the Rhineland Study, access can be obtained by submitting a formal request to the Data Access Committee (RS-DUAC; rs-duac@dzne.de) for academic, non-commercial research purposes, and in accordance with the Rhineland Study's Data Use and Access Policy (https://www.rheinland-studie.de/en/participation/rules-of-use/). The committee will review all requests within 4 weeks. Access may be subject to a completed data transfer agreement. Similarly, information on data access and contact details for the EPIC-Potsdam study can be obtained at https://www.dife.de/en/research/cooperations/epic-study/. Access can be obtained for academic, non-commercial research purposes by submitting a formal request to the Principal Investigator of the EPIC-Potsdam Study (Prof. Dr. Matthias Schulze; mschulze@dife.de). The request will be reviewed within 4 weeks, and access may be subject to data use agreements. All authors had full access to the data and take responsibility for the integrity of the data as well as the accuracy of the analysis. Although the raw individual-level data are protected and not available due to privacy laws, summary statistics of the meta-analysis GWAS for model 1 and model 2 are publicly available via the GWAS catalog under accession number GCP001611.

## Code availability

All custom code and scripts used in the present study are publicly available through Zenodo:https://doi.org/10.5281/zenodo.19054283.

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

## Acknowledgments

We would like to thank all participants and the study personnel of the Rhineland Study and EPIC-Potsdam. **MMBB** discloses support for the research of this work from the Diet-Body-Brain Competence Cluster in Nutrition Research funded by the Federal Ministry of Education and Research (grant number 01EA1410C), the Federal Ministry of Education and Research (FKZ:01KX2230) in the framework "PreBeDem - Mit Prävention und Behandlung gegen Demenz", the Deutsche Forschungsgemeinschaft (DFG, German Research Foundation) under Germany´s Excellence Strategy – EXC 2151 – 390873048, the Deutsche Forschungsgemeinschaft (DFG, German Research Foundation) – SFB 1454 – Project-ID 432325352, and the Helmholtz Association under the 2023 Innovation Pool. **NAA** is supported by the European Research Council Starting Grant (Number: 101041677). **MBS** is supported by a grant from the European Commission and the German Federal Ministry of Education and Research within the Joint Programming Initiative A Healthy Diet for a Healthy Life, within the ERA-HDHL cofounded joint call Biomarkers for Nutrition and Health (01EA1704), grants from the German Federal Ministry of Education and Research and the State of Brandenburg to the German Center for Diabetes Research (DZD) (82DZD00302, 82DZD03D03). **FE** is supported by grants from the German Federal Ministry of Education and Research and the State of Brandenburg to the German Center for Diabetes Research (DZD) (82DZD00302, 82DZD03D03). **ENL**, **MAI**, and **VT** declare no relevant funding.

## Author contributions

E.N.L.**:** Conceptualization, Methodology, Formal Analysis, Writing—Original Draft Preparation, Visualization; M.A.I.**:** Conceptualization, Methodology, Formal Analysis, Writing—Original Draft Preparation, Visualization; V.T.**:** Conceptualization, Methodology, Formal Analysis, Writing—Original Draft Preparation, Visualization; F.E.**:** Data Curation, Formal Analysis, Writing—Reviewing and Editing; M.B.S.**:** Data Curation, Resources, Writing—Reviewing and Editing; N.A.A.**:** Conceptualization, Methodology, Writing—Reviewing and Editing, Data Curation, Supervision; M.M.B.B.**:** Conceptualization, Methodology, Resources, Writing—Reviewing and Editing, Data Curation, Funding Acquisition, Supervision.

## Funding

## Competing interests

E.N. Landstra, M.A. Imtiaz, V. Talevi, F. Eichelmann, M.B. Schulze, N.A. Aziz and M.M.B. Breteler report no competing interests relevant to this work.
