## [Transparent Peer Review file · Nature Communications]

Population-based Genome-wide Association Study of Plasma Complex Lipid Species

Corresponding Author: Professor Monique Breteler

Version 0:

Reviewer comments:

Reviewer #1

(Remarks to the Author)

Landstra, Imtiaz and colleagues report a large GWAS on complex lipids, which includes close to 1000 lipids plus composite measures in around 6000 participants of the Rhineland study. The study is well powered and includes two replication datasets. The dataset is large and highly valuable, and the authors claim to have found several novel loci and associations, which they further analyzed in Mendelian randomization experiments against eQTLs, mediation analysis and with respect to previously reported trait associations. While the work certainly has merit and adds to the field, there are several major concerns that need to be addressed:

1. The authors performed both replication analyses and checks against previous work to annotate lipid-associated loci. However, it seems that the comparison with the study by Cadby, Giles et al., Nat Communications, 2022 is incomplete. For instance, NIPAL1 is here reported as a novel locus associated with an HcEr (number 15 in STable 1), including SNP rs3805187, but that SNP is also reported in the previous study by Cadby, Giles et al., also with HcEr species and a minimum P-value = $1e-12$ (see https://metabolomics.baker.edu.au/pheweb_standard/variant/4:48076562-T-A). Similarly, rs1260326 (annotated to GCKR) is listed in Supplementary Table 8 as having no associations in the Cadby study, but shows significant associations there. A correct annotation of loci as known vs. novel will benefit from a more thorough comparison here.
2. In general, the annotation of known vs. novel loci is inconsistent comparing previous literature and also the data provided in the Supplement. For instance, ABCA7 is reported as novel locus in the main text, but is not marked as novel in Supplement Table 1. The supplement is correct, as the locus has been reported before, so the manuscript should be carefully revised in that regard. Another example is TRIM48 (locus 30, lead SNP rs1589680), which is not marked as novel in STable 1 but not annotated as previously reported in STable 8.
3. Further, it seems that several of the loci reported to be novel fail to replicate in the other studies. An example is locus 4 (rs634691 mapped to USH2A). The annotation provided does not allow further interpretation, as it is not mentioned how many SNPs in the locus associate with lipids (if it were only one SNP, chances are high that this is a false positive) or if the SNP was imputed. Supplements should be updated to include such information and it should be critically discussed in the main text how confident the authors are in such loci. Other examples, such as the aforementioned TRIM48 locus (and other similar ones) are raising additional concerns, as associations with $P < 1e-200$ are expected to replicate more broadly than with a few nominally significant associations in the replication study/the Cadby study.
4. There is no information on fasting status in the Rhineland study. As fasting/non-fasting status has severe consequences on the lipid profile, this information needs to be added to the manuscript. If there is a mix of fasting vs. non-fasting samples, association tests should be adjusted for fasting status.
5. Transcriptomics data generation and preprocessing in the Rhineland study is not described and needs to be added to the methods section in sufficient detail. The manuscript does also not discuss why a) GTEx was chosen for 2-sample MR over a quite well powered 1-sample MR analysis in the Rhineland study itself; and b) why mediation analysis was chosen over 1-sample MR to assess causal effects on gene expression.

6. For the MR results, horizontal pleiotropy has seemingly not been assessed for all tested genetic instruments. The MR-Egger intercept should be reported for all reported results and an additional test, such as in MR-PRESSO (Verbanck et al., Nature Genetics, 2018, <https://doi.org/10.1038/s41588-018-0099-7>), would be advisable.

7. PheWAS with simple lookups of overlapping associations are not considered state-of-the-art anymore. A thorough assessment through colocalization is required to draw any conclusions. Methods for summary statistic-based colocalization analyses like HyPrColoc (Foley et al., Nature Communications, 2021, <https://doi.org/10.1038/s41467-020-20885-8>) enable such analyses without the need for access to individual-level data.

Minor comments:

1. Figures seem to be screenshots, not high-resolution pictures. Axis limits for Figure 3c and d should be fixed to make the figures more readable. Data points in Figure 3b should be annotated with error confidence intervals if possible.

2. The discussion reads very superficially, can be condensed and should be strengthened by discussing in more detail why and how this study adds to the existing literature beyond simply listing associations. E.g., the paragraph on PheWAS results for CPS1, PKD2L (which, based on extended tables should be correctly labeled PKD2L1), and PNPLA3 is rather uninformative, as the functions of those genes are not described nor are they linked to the observed associations. While the associations for PNPLA3 are plausible given the gene's function, this is not provided for CPS1 or PKD2L1.

3. Beyond simple replication analysis, the authors could consider performing a meta-analysis for overlapping data in the EPIC-Potsdam study. This is a missed opportunity that could be quickly addressed.

Reviewer #2

(Remarks to the Author)

Authors present a GWAS of 1000+ lipid traits categorised into two broad types – 970 lipid species and 267 fatty acid composites. Primary/discovery analyses were run in the Rhineland cohort (n=6096) and replication/validation for a subset of lipid traits performed in two independent cohorts (FinnGen, n=7266; EPIC-Potsdam, n=1188). In addition, follow-up analyses were done including PheWAS, two-sample MR (using eQTLs) and gene expression analysis (including mediation). This study has advanced the field by identifying 30 novel genomic loci associated with lipid species, and 25 novel genomic loci associated with fatty acid (FA) composite measures.

Overall, I found this to be a well-presented GWAS the novelty of which comes from the range of lipid traits studied. Compared to established literature, this study uses a more comprehensive approach to lipidomic profiling by evaluated the genetic architecture of the whole plasma FA composition as well as the shared biochemical properties across different lipid species and classes, including their tail length and degree of saturation. However, to maximise the value of this contribution to the literature authors should ensure their reporting meets the relevant community-agreed standards. Related to this, I strongly encourage authors to follow and report against the STREGA checklist for GWAS (<https://www.equator-network.org/reporting-guidelines/strobe-strega/>), the STROBE checklist for observational analyses (<https://www.equator-network.org/reporting-guidelines/strobe/>) and the STROBE-MR checklist for MR (<https://www.equator-network.org/reporting-guidelines/strobe-mr-statement/>). Also related to this, authors should consider use of the LSI Lipidomics Minimal Reporting Checklist (https://lipidomicstandards.org/reporting_checklist/).

Furthermore, analytical code should be made openly available in Github (or similar) and all GWAS summary statistics made openly available.

In general, the methodology appears robust and follows established quality control and methodology standards in GWAS and lipidomic studies. However, some specific queries/points for clarification are listed below.

Major comments:

1. Methods: currently all analyses are adjusted for sex giving sex-averaged results. Given the likely heterogeneity in lipid concentrations and their genetic contributions across males and females, a very useful addition would be to see results of GWAS conducted separately in these subgroups.

2. Results, p.6, heritability estimation:

a. Providing 95% CI for the h^2 estimates would be more informative than SE.

b. Estimates of $h^2=1$ seem inconceivable – isn't it more likely there is an issue with the model convergence and/or assumptions here? Based on Supp Figure 2a – these are outlying values. Could differential LD between associated and not associated regions of the genome be an issue? Also, an $h^2=1$ with a wide 95% CI isn't necessarily meaningful. I suggest these results are checked for robustness as well as greater caution in their reporting.

c. Can you explain the rationale for adjusting for the clinical lipid measurements in the h^2 analysis. It doesn't make sense to me to adjust for heritable traits in this way as then you're adjusting for these at the phenotype level (i.e., accounting for genetic and non-genetic variance). I think this could have unintended consequences on resulting estimates (e.g. bias). For me this makes interpretation of these results challenging.

3. Results, p8: regarding the validation analyses

a. Reporting overall concordance across all three studies would be most informative. Looking at concordance of betas as well as p-values would help to establish whether the relative lack of replication in EPIC-Potsdam is related to power or genuine heterogeneity in signal.

- b. Could you also compare results with: <https://www.nature.com/articles/s41467-022-30875-7> (where traits overlap)?
4. Results: the inclusion of some heatmaps or similar to show the phenotypic correlation across traits would be useful.
5. Results, p9: regarding the MR analysis
- What correction was made for multiple testing?
 - There is a need to comment on the potential bias in the causal estimate due to pleiotropy, especially given that some of the MR-Egger intercepts indicate the presence of pleiotropy. In addition, there are many genes where no MR-Egger sensitivity test could be performed and therefore the potential bias due to pleiotropy is unknown. Instrumenting correlated traits like lipids is hard to do – this should be acknowledged.
6. Results: How was it determined that loci were ‘novel’ (i.e., what criteria were used)?
7. Methods, p13, GWAS models: The manuscript could benefit from further clarification regarding the implications of adjusting for clinical lipid measurements in model comparisons.
8. Methods, p15, two-sample MR:
- How was the adjustment for the traditional lipid measures done? In two-sample MR, the same covars need to be adjusted for in both the SNP-exposure and SNP-outcome GWAS, especially in the case of covars with a heritable component.
 - This doesn’t make sense to me: “Specifically, eQTLs of candidate genes were used as the exposure and variants associated with lipid species levels as the outcome.” Instruments (associated SNPs) are selected in the exposure GWAS not the outcome GWAS?
 - Given you have expression data in the cohort, could you do a one-sample MR?
9. Methods, p15, mediation analysis: please could you provide more detail of the approach and model used for the mediation analysis. It’s not obvious how the ‘lavaan’ package would have been used for mediation? Nor do I understand what the mediation ‘estimate’ in Table 2 is or how it relates to mediation results in Suppl Data 14. Providing the code (as mentioned above) could help clarify.
10. Discussion, p11: here it says, “Results of our PheWAS further highlight the important role of complex lipids as modifiers of a large number of different traits and diseases.” However, there has been no mention of the potential bias due to reverse causality here, that the disease phenotype may be driving changes in lipid metabolism and therefore the lipid profile and FA composition. This need addressing in the discussion.

Minor comments:

11. Results, p4: I don’t completely understand the approach to defining loci and lead SNPs as described here (“These genomic loci were identified ... significant SNPs”) and in the corresponding section of the Methods (p13). Can you reword and/or add a diagram (in suppl) to help?
12. Results, p.4: define what the % in the brackets represents in the sentence: “The lipid species within the sphingomyelins (SMs) (100%)”
13. Several of the Supplementary Data Tables (in the excel file) need additional annotation to help interpretation. For example, in sp1, it’s not clear to me what the three results ‘chunks’ represent, save for the fact one of them seems to be from FinnGen. Same for sp3 & sp4.
14. Results, figure 3: I don’t find any elements of this figure particularly informative – in particular (b). For a, c, d, would boxplots showing the distribution of correlation coefficient by class be more effective?! Same applies to suppl figure 1.
15. Results, supp data 5: shading at the legend at the top is suggested but seemingly not applied. Columns J & U seem to be indicating which model the SNP/lipid association comes from but what about where they appear in both models?
16. Results, Supp Figure 2a: this would be easier to follow if the lipid classes were presented in the same order in the two plots or if the two plots were combined into a two-way boxplot where boxes for models 1 & 2 for the same lipid were next to each other.
17. Results, p7: in the top paragraph where it says: “After additional adjustment for clinical lipid measurements ...” – I don’t understand what analysis this (and the rest of this paragraph) is referring to because I thought the whole paragraph was about combined model 1 and 2 results.
18. Results, p7: re the GTEx analysis – please specify tissue type.
19. Results, p.7, second paragraph: I’m not what analysis the final sentence of this paragraph is related to (“Additionally, we identified 64 and 82 mQTLs ...”) – it doesn’t seem to fit with the rest of the paragraph which is about GTEx data and eQTLs.
20. Results, supp data 12 – what does the highlighting indicate? Add key/legend.
21. Results, p8: check reference to supp data 1 at the top of this page – is this correct?
22. Results, suppl data 10: could you add the associated lipid traits to this table?
23. Results, suppl data 14: further explanation of the results here is needed. Is this all based on the observational analysis of expression data? And should there be a gap between the tables in columns A-H and I-Q or is this a continuation in which case what’s the relation between the genes in the first part of the table (A-H) and the genes/SNPs in the second part (I-Q)?
24. Results, p9: not sure how ‘top modules’ have been defined – top based on what?
25. Method, p13: what is the rationale behind performing a transformation on the data (residuals) after having fitted a linear model (that assumes a normal distribution) – if the raw lipid data were not normally distributed, shouldn’t they be transformed prior to regressing out the fixed effects?
26. Method, p.13: Can the authors comment on use of the somewhat lenient info score threshold of 0.3 – why was this cut off chosen and what are the implications?
27. Discussion, p12: worth mentioning that a limitation of the validation analysis is that many associations could not be tested in either cohort.
28. Discussion: Possible impact of variations in lipid measurement techniques across cohorts on validation/concordance with literature should be discussed.
29. Discussion: Clinical Relevance: A deeper discussion on how the identified loci might inform clinical interventions or risk stratification would strengthen the manuscript’s translational relevance.

Reviewer #3

(Remarks to the Author)

Version 1:

Reviewer comments:

Reviewer #1

(Remarks to the Author)

I appreciate the extensive work that the authors have put into this revision.

However, based on the additional data provided, I still have major concerns and must get back to the third point I raised in my previous review. There, I particularly questioned the reliability of loci defined by a single SNP association, which are furthermore not replicated in any of the other studies investigated by the authors.

While reviewing the regional association plots provided in this revision, I was surprised by the large number of single SNP associations reported. If there is only one SNP found to be associated, but the other SNPs in the locus are not showing any association (or weaker/insignificant association but no LD), it is highly likely that this is a false positive. This is even more likely in cases where one SNP shows an association and other SNPs in LD do not even show a trend towards association. This becomes an issue, as I counted around 20 such cases in both the model 1 and model 2 discovery studies, which is a third of all associations. Even more concerning is that many of these are loci reported to be novel.

This observation casts doubt on the correctness of the reported results, which led me to investigate all parts and data provided in the revised version in detail. This is time-consuming, and will take more time. However, this issue - for me - must be resolved, and can be resolved while I continue to review other aspects of this work. I should note in this regard that I personally see the data used and the study by itself as valuable and impactful, and that this assessment will not change because fewer loci are reported. But the reported findings should be correct to the best knowledge of all people involved.

I hence urge the authors to either revisit their methodological approach or at least provide the following information in addition for all associations:

- a) the number of significant SNPs per association in a locus (you now provided the number of SNPs in LD, which by itself is not sufficient).
- b) the information if significant SNPs were imputed or genotyped directly. If imputed, the info score should also be provided.

I was further asked by the editor to particularly review the methods for generating and processing the lipidomics data. There are two aspects that raise concerns:

- a) Lipidomics data is often log-normally distributed, which requires log-transformation even if data is quantitative. The manuscript does not state transformation of lipid concentrations. I would advise to investigate if the distributions of lipid concentrations are skewed or normal. If they are (close to) normally distributed, this should be stated. Otherwise, I would advise to perform log-transformation.
- b) Inclusion of lipids present in at least 1% of the population is a very relaxed threshold. Based on the total number of participants, this includes GWASs with around 60 individuals. Aside from low power, a high amount of missing data in lipid measurements can point to issues in measurement reliability. I would ask the authors to revisit this approach, or at least provide some rationale on why they used this threshold.

I apologize for missing this in my first review.

Reviewer #2

(Remarks to the Author)

This manuscript has improved substantially following review and I thank the authors for their comprehensive response to my previous comments. Below are some outstanding points for consideration.

Major:

1. As results of your GWAS suggest, many mQTL are non-specific in that they show associations across a range of lipid traits. As such, in many cases (where you perform 2-sample MR with lipid traits as exposures) it is unrealistic to expect to be able to instrument single lipid traits in isolation. That is to say, even where you have a seemingly robust set of instruments for lipid trait 'a', those same SNPs may well proxy other lipid traits (both measured and unmeasured) in the dataset (i.e. they are shared instruments). I think this point is at the very least worth acknowledging in the discussion.

Minor:

2. I couldn't see that the abbreviations in Suppl. Fig 1 were defined anywhere.
3. I couldn't see a reference to Supplementary data 9 & 13 (apologies if I missed it).
4. "In addition to the eQTLs, we also identified 65 and 80 mQTLs which were associated with the lipid species in model 1 and 2, respectively, which were also detected in previous lipidomic GWAS studies (26)." I still think this sentence is in the wrong place. And I don't understand why you are singling out the '65 and 80 mQTLs' when the whole analysis is about identifying mQTLs.
5. L257, should this be referencing Suppl Fig 6a (not 3a)?
6. L302 – check this reference to Suppl Data 15 – not sure it is correct.
7. Still some of the suppl data tables could do with additional column headers and/or labels – eg Supp data 16 – different betas need explanations, supp data 18 also.
8. L538-540: repeated sentence.
9. For clarity, in the GWAS, did you analyse dosages/probabilities/bestguess genotypes?

Version 2:

Reviewer comments:

Reviewer #2

(Remarks to the Author)

I am satisfied with the authors responses to all my queries and have no further comments.

Reviewer #4

(Remarks to the Author)

The manuscript has been substantially improved in terms of transparency, particularly with the addition of imputation quality metrics, genotyping status, and per-lipid missingness summaries. The transformation strategy and primary GWAS quality control procedures are generally appropriate.

However, two related issues warrant further clarification. First, several loci appear to be defined by a single associated variant without clear LD-supported regional signal. While the revised tables improve reporting, additional analyses demonstrating the robustness of these loci (e.g., sensitivity to stricter imputation quality thresholds, minor allele count filters, and/or cohort-specific consistency) would strengthen confidence in these findings.

Second, although most lipid species were measured in several thousand individuals, a subset shows substantially lower effective sample sizes. It would be helpful to clarify whether a minimum per-trait sample size threshold was applied for GWAS inclusion and how associations from lower-N traits were handled in interpretation.

Addressing these points would substantially reinforce the robustness of the reported novel loci.

Reviewer #5

(Remarks to the Author)

I am a new reviewer and was asked to address responsiveness of the authors to Reviewer 1's comments. I have read the manuscript, the response to reviewers and reviewed the relevant Supplemental Tables.

I find the response to all reviewer comments to be appropriate and satisfactory.

Small note on the first comment, the authors sometimes list "1" as the number of GWASsnps and nsnp in Supp Table 1, but then the authors themselves list several different imputed and genotyped variants in the locus (e.g. locus 29). I found this confusing. It might be nsnp per trait, but I don't think that would be 1 in most cases either. You might want to correct the numbers in that column to reflect the number of variants in the locus (they do not necessarily have to be in high LD).

REVIEWER COMMENTS

Reviewer #1 (Remarks to the Author):

Landstra, Imtiaz and colleagues report a large GWAS on complex lipids, which includes close to 1000 lipids plus composite measures in around 6000 participants of the Rhineland study. The study is well powered and includes two replication datasets. The dataset is large and highly valuable, and the authors claim to have found several novel loci and associations, which they further analyzed in Mendelian randomization experiments against eQTLs, mediation analysis and with respect to previously reported trait associations. While the work certainly has merit and adds to the field, there are several major concerns that need to be addressed:

1. The authors performed both replication analyses and checks against previous work to annotate lipid-associated loci. However, it seems that the comparison with the study by Cadby, Giles et al., Nat Communications, 2022 is incomplete. For instance, NIPAL1 is here reported as a novel locus associated with an HCer (number 15 in STable 1), including SNP rs3805187, but that SNP is also reported in the previous study by Cadby, Giles et al., also with HCer species and a minimum P-value = $1e-12$ (see https://metabolomics.baker.edu.au/pheweb_standard/variant/4:48076562-T-A). Similarly, rs1260326 (annotated to GCKR) is listed in Supplementary Table 8 as having no associations in the Cadby study, but shows significant associations there. A correct annotation of loci as known vs. novel will benefit from a more thorough comparison here.

Response to Point 1:

We thank the reviewer for highlighting this important point. Accordingly, we have double-checked all our findings against those reported previously by Cadby et al., Ottensmann et al. and mGWAS-Explorer (<https://www.mgwas.ca/mGWAS/upload/SNPUploadView.xhtml>). Specifically, we compared our results from model 1 with these previously published GWAS summary statistics. This slightly reduced the number of novel loci in our paper to 25. We have marked the known and novel loci using different colors in the tables (**Supplementary Tables 1 to 7**). We also rechecked the tables and ensured that all comparisons are correct. In some cases, there might be an overlap between species, as the study by Cadby et al. only included lipid summary measures (e.g. TAG53:6) but no further details on fatty acid tails that would allow for a precise match with our data (e.g. TAG53:6(FA18:2)). They might also not have the exact same structure, such as our measured CER(24:1) versus CER(d14:1/24:1)). Similarly, because the association of the *NIPAL1* locus with HCER (FA22:1) was not reported previously, we considered this locus to be novel for this lipid species. Thus, to avoid ambiguous comparisons, these types of uncertain matches were not included.

We also checked whether exact lipid species-lead SNP associations at each locus (**Supplementary table 1-4,6,7**) identified in the Rhineland Study replicated in the previous studies (Cadby et al & Ottensmann et al).

Novel mQTLs (**Supplementary table 10-12**) were defined as those SNPs that had not previously been associated with any lipid trait or species.

2. In general, the annotation of known vs. novel loci is inconsistent comparing previous literature and also the data provided in the Supplement. For instance, ABCA7 is reported as novel locus in the main text, but is not marked as novel in Supplement Table 1. The supplement is correct, as the locus has been reported before, so the manuscript should be carefully revised in that regard. Another example is TRIM48 (locus 30, lead SNP rs1589680), which is not marked as novel in STable 1 but not annotated as previously reported in STable

Response to Point 2:

Although the *ABCA7* locus was associated with Hex2Cer species in the study by Cadby et al., its association with the LCER lipid species, which were not available in the previous study, is novel (**Supplementary table 1-4, 6, 7**).

On the other hand, as mentioned above, novel mQTLs were defined as those SNPs that had not previously been associated with *any* lipid trait or species. Therefore, given that the mQTLs at this locus (rs4147909 and rs4147929) had previously been reported to be associated with other lipid species (Cadby et al and <https://www.mgwas.ca/mGWAS/upload/SNPUploadView.xhtml>), these were not marked as novel (**Supplementary table 10-12**).

We also corrected the annotation for the *TRIM48* locus as it is indeed a novel locus in our study. We have rechecked and redone all the Tables to ensure that everything is correct.

3. Further, it seems that several of the loci reported to be novel fail to replicate in the other studies. An example is locus 4 (rs634691 mapped to *USH2A*). The annotation provided does not allow further interpretation, as it is not mentioned how many SNPs in the locus associate with lipids (if it were only one SNP, chances are high that this is a false positive) or if the SNP was imputed. Supplements should be updated to include such information and it should be critically discussed in the main text how confident the authors are in such loci. Other examples, such as the aforementioned *TRIM48* locus (and other similar ones) are raising additional concerns, as associations with $P < 1e-200$ are expected to replicate more broadly than with a few nominally significant associations in the replication study/the Cadby study.

Response to Point 3:

We thank the reviewer for raising this issue. The specific SNP mentioned by the reviewer (rs634691) mapped to *USH2A* in our study and was related to LPCs (LPC(20:5) and LPC(18:3)). The SNP rs634691 was the lead SNP at this locus (see plot below). This locus plot suggests that the association is not robust, which may account for the fact that this locus was not replicated in the EPIC Potsdam cohort. Nevertheless, the *USH2A* locus has previously also been associated with lipid species (<https://www.ebi.ac.uk/gwas/genes/USH2A>, <https://www.nature.com/articles/s41598-019-56496-7>).

The reviewer is correct to point out that the *TRIM48* locus (tagged by rs1589680), which was strongly associated with several TG species in the Rhineland Study (see plot below), did not replicate in the EPIC Potsdam cohort. One potential explanation could be the substantially lower sample size of the replication cohort ($n=1,888$). This locus was also not reported in the Cadby et al. GWAS for the SNPs in LD (**Supplementary Table 1**).

To further clarify the broader issue the reviewer mentioned, as well as to enable direct assessment of the evidence regarding the locus being associated with lipids, we have now also included the number of SNPs in LD as an additional column in each supplementary table. Moreover, we have now also included the locus plot for each locus as supplementary Figure 2 “Locus zoom plots for genomic loci identified in model 1 (A) and model 2 (B).” to allow for locus-specific visual evaluation.

4. There is no information on fasting status in the Rhineland study. As fasting/non-fasting status has severe consequences on the lipid profile, this information needs to be added to the manuscript. If there is a mix of fasting vs. non-fasting samples, association tests should be adjusted for fasting status.

Response to point 4:

Only very few blood samples were collected in a non-fasting state in the Rhineland Study (n= 51). We have now clarified this in the methods section of the revised version of the manuscript. Moreover, as the reviewer

suggested, we have now additionally adjusted for the potential effect of fasting status in model 2, which we have now elaborated upon in the methods section as well.

“Absolute (nmol/ml) plasma lipid concentrations were measured using the Metabolon Complex Lipids platform. Almost all blood samples were collected after an overnight fast (fasting: n= 6045 fasting, non-fasting: n= 51).”

“In the second model, we additionally adjusted for fasting status, as well as the levels of total cholesterol, triglycerides, HDL-C, and the use of lipid-lowering medication (model 2).”

As expected, based on the small number of participants who donated non-fasting blood samples, further adjustment for fasting status did not materially change any of our findings.

5. Transcriptomics data generation and preprocessing in the Rhineland study is not described and needs to be added to the methods section in sufficient detail. The manuscript does also not discuss why a) GTEx was chosen for 2-sample MR over a quite well powered 1-sample MR analysis in the Rhineland study itself; and b) why mediation analysis was chosen over 1-sample MR to assess causal effects on gene expression.

Response to point 5:

We would like to thank the reviewer for this comment. The details of transcriptomics data generation and preprocessing were already included in the manuscript (previously at line 586). However, to improve clarity and maintain consistency with the flow of the Methods section, we have now relocated this section to follow the “Lipidomics” and “Genomics” paragraphs (line 523).

Following the suggestion made by the reviewer, we also performed one-sample MR analyses as follows (Methods section):

“We employed a one-sample MR approach to test for evidence that the associations between genes identified through our top-down and bottom-up approach and lipid species (adjusted for clinical lipid measures) were causal, using the OneSampleMR package (version 0.1.5) (64). To this end, gene expression levels were first adjusted for age, sex, batch effects, red and white blood cell counts, and the fractions of basophils, eosinophils, lymphocytes, monocytes, and neutrophils, and residuals were normalized using a rank-based inverse normal transformation. Subsequently, for each gene of interest (exposure), we run a GWAS to identify genome-wide significant SNPs associated with its expression levels that could serve as genetic instruments. Lipid species were treated as the outcome. We clumped SNPs in LD ($r^2 < 0.01$ within a 10 Mb window). Prior to analysis, lipid species levels were adjusted for age, sex, total cholesterol, triglycerides, HDL-C, lipid-lowering medication use, fasting status, and the first 10 genetic PCs. The residuals from this model were also normalized using a rank-based inverse normal transformation. The MR analysis was then performed using two-stage least squares (2SLS) instrumental variable regression. In this framework, the outcome (lipid species) is regressed on covariates and gene expression levels, while gene expression is instrumented by the selected SNPs. This approach allowed us to estimate the causal effect of gene expression on lipid species, while accounting for potential confounding. We utilized the F-statistic as a measure of the strength of SNP-instruments, the Wu-Hausman test p-values for assessing endogeneity, and the Sargan test p-values for checking instrument validity, ensuring that all model assumptions were met. We set the statistical significance level at $FDR < 0.05$ for identifying significant causal associations.”

In parallel we also run two-sample MR analyses, which are less prone to overfitting, to assess the robustness of our one-sample MR results, as follows (Methods section):

“In complementary analyses, we also employed a two-sample MR approach to assess the robustness of the potentially causal associations between the genes and lipid species identified in our one-sample MR analyses, using the TwoSample MR package, because compared to one-sample MR, two-sample MR is less prone to overfitting. We downloaded whole blood eQTL data from the GTEx resource (65), and used variants associated with gene expression levels as genetic instruments in the MR analyses. Specifically, eQTLs of candidate genes were used as the exposure and variants associated with lipid species levels as the outcome. We clumped SNPs in LD ($r^2 < 0.01$ within a 10 Mb window). The method selection for Two-sample MR was based on the number of

SNP instruments available, including Wald Ratio (1 SNP instrument), inverse variance weighted method (≥ 2 SNP instruments), and MR-Egger method (> 3 SNP instruments). To assess the robustness of our MR results, we performed several sensitivity analyses, such as calculating F-statistics to assess the risk of weak-instrumental bias and MR-Egger intercept to identify horizontal pleiotropy.”

All results of the MR analyses are included in **Supplementary table 16**. In addition, we also opted to keep the functional mediation analyses, which allow for the direct estimation of the *proportion* of the effect mediated through gene expression.

6. For the MR results, horizontal pleiotropy has seemingly not been assessed for all tested genetic instruments. The MR-Egger intercept should be reported for all reported results and an additional test, such as in MR-PRESSO (Verbanck et al., Nature Genetics, 2018, <https://doi.org/10.1038/s41588-018-0099-7>), would be advisable.

Response to point 6:

We thank the reviewer for this helpful suggestion. We evaluated horizontal pleiotropy by assessing the MR-Egger intercepts for all tested genetic instruments (all MR-Egger intercepts are included in **Supplementary Table 16**). Additionally, we used MR-PRESSO to assess pleiotropy for lipid species associations with diseases, and vice versa, when looking for MR-based associations of lipid species with diseases (**Supplementary table 20**).

7. PheWAS with simple lookups of overlapping associations are not considered state-of-the-art anymore. A thorough assessment through colocalization is required to draw any conclusions. Methods for summary statistic-based colocalization analyses like HyPrColoc (Foley et al., Nature Communications, 2021, <https://doi.org/10.1038/s41467-020-20885-8>) enable such analyses without the need for access to individual-level data.

Response to point 7:

We agree with the reviewer and have now also included colocalization analyses using the COLOC package. The results are reported in the Results section of the manuscript under the heading “*Colocalization and two-sample Mendelian Randomization analyses*”, as follows:

“Guided by the results of our PheWASs, we performed additional colocalization analyses for lipid species-associated genetic variants mapping to the ZPR1, APOE and SUGP1 loci and coronary artery disease (27), those mapping to PNPLA3, SUGP1 and ZPR1 loci and type 2 diabetes (28), those mapping to APOE and ABCA7 loci and Alzheimer’s disease (29), and those mapping to FADS2, PKD2L1 and CPS1 loci and chronic kidney disease (30). Colocalization analyses revealed that the lipid species-associated genetic variants within the ZPR1, APOE and SUGP1 loci and coronary artery disease, those within the SUGP1 locus and type 2 diabetes, and those within the CPS1 locus and chronic kidney disease, shared a single causal variant with posterior probabilities of the single shared causal variant ($PP:H4 > 0.99$ (**Supplementary Data 19**)).”

The description of the approach is included in the Methods section of the paper under the heading

“*Colocalization and two-sample Mendelian Randomization Analyses*” as follows:

“We performed colocalization analyses to determine whether the genetic variants associated with lipid species overlapped with those related to disease traits identified in our PheWAS analyses. For each lipid trait, we considered variants within ± 500 kb of the lead SNPs and merged them with summary statistics from publicly available GWAS data on coronary artery disease, Alzheimer’s disease, chronic kidney disease and type 2 diabetes. Colocalization was conducted using the *coloc.abf* function from the *coloc* R package (62), which estimates posterior probabilities for five hypotheses: (H0) neither trait is associated with the region; (H1/H2) only one trait is associated; (H3) both traits are associated but with distinct causal variants; and (H4) both traits are associated and share a single causal variant. The prior probability for a shared causal variant (H4) was set at $1.0E10^{-6}$. Posterior probabilities were summarized across all lipid traits to identify top colocalizing regions, with particular attention to traits showing strong evidence of colocalization (e.g., $PP.H4 > 0.9$).”

Minor comments:

1. Figures seem to be screenshots, not high-resolution pictures. Axis limits for Figure 3c and d should be fixed to make the figures more readable. Data points in Figure 3b should be annotated with error confidence intervals if possible.

In accord with the reviewer's helpful suggestions, we have now modified the beta-beta plots, using high-resolution pictures and correcting the axis limits for clarity. Some of the previous subplots (i.e., Figure 3b and c) are now included in Figure 3a.

Figure 3a:

Data points in Figure 3b are additionally annotated with error confidence intervals and limited to the top 30 lipid species (based on genetic correlation p-values). The depictions of 95% confidence intervals are capped at -2 and 2 .

Genetic Correlation: HDL-C

Genetic Correlation: LDL-C

2. The discussion reads very superficially, can be condensed and should be strengthened by discussing in more detail why and how this study adds to the existing literature beyond simply listing associations. E.g., the paragraph on PheWAS results for CPS1, PKD2L (which, based on extended tables should be correctly labeled PKD2L1), and PNPLA3 is rather uninformative, as the functions of those genes are not described nor are they linked to the observed associations. While the associations for PNPLA3 are plausible given the gene’s function, this is not provided for CPS1 or PKD2L1.

We have provided further detail on different sections in the discussion section:

“Results of our PheWAS further highlight the important role of complex lipids as modifiers of a large number of different traits and diseases. For example, genetic variants in CPS1 have previously been associated with a variety of different phenotypes, including HDL-C levels and diabetes (40). The rare allele of the CPS1 variant was previously related to a lower risk of coronary artery disease (CAD), specifically in women (41). We found that CPS1 was specific to monounsaturated cholesteryl esters, which are intimately linked to HDL-C as well as chronic kidney disease, diabetes and other diseases (13,42), and have previously been shown to have sex-specific effects (43). Therefore, this link between CPS1 and cholesteryl esters could explain previous findings on the risk of CAD. Another hub gene, PKD2L, regulates a crucial membrane protein and has previously been associated with cardiometabolic health as well as phospholipid measurements among others. Although the mechanism linking this gene to lipids is still unclear, it has previously been linked to phospholipid measurements as well. We further extend these findings by showing that PKD2L mainly affects monounsaturated lipids and could be crucial for lipid metabolism, possibly through membrane interactions as lipids are crucial parts of membranes as well.”

We have also highlighted potential clinical and translational interpretations:

“These results could pave the way to precision medicine and improved risk stratification. For example, measuring the identified lipids could not only aid in identifying individuals with a risk allele – without complex genetic analyses –, but could possibly predict disease better than the traditional lipid measurements in disease risk models. Furthermore, by identifying individuals with a genetic risk factor, stricter monitoring and regulating of these identified lipids could possibly aid in preventing disease. Therewith, it could improve precision medicine. In addition to the possible gene-lipid-disease links,...”

3. Beyond simple replication analysis, the authors could consider performing a meta-analysis for overlapping data in the EPIC-Potsdam study. This is a missed opportunity that could be quickly addressed.

We have now also performed a meta-analysis for overlapping data for both our base (model 1) and fully adjusted (model 2) models. A total of 64 and 68 genomic loci were identified for model 1 and model 2, respectively, of which 31 and 41 were novel. The results are included under the heading “*Meta-analysis of results*” in the *Results* section of the manuscript, as follows:

“To integrate all findings, we meta-analysed the results from the Rhineland Study and EPIC-Potsdam. After adjustment for sex, age, and the first 10 genetic principal components (model 1), followed by adjustment for HDL-C, total serum triglycerides, total cholesterol levels, use of lipid-lowering medication and fasting status (model 2), we found a total of 64 genomic loci, 31 of which were novel in model 1, and 68 genomic loci, 41 of which are novel in model 2. Overall, meta-analysis results were similar to the results from the Rhineland Study, except for the identification of 10 extra genomic loci in model 1 (PAQR9, MLXIPL, SLC05A1, TTC39B, SLC22A24, MOGAT2, M6PR, SOAT2, LIPG, GRAMD4) and 9 extra loci (SLC44A1, UGCG, C9orf91, C10orf82, SLC22A24, SEC14L5, LIPG, ATP8B1 and GRAMD4) in model 2.”

Reviewer #2 (Remarks to the Author):

Authors present a GWAS of 1000+ lipid traits categorised into two broad types – 970 lipid species and 267 fatty acid composites. Primary/discovery analyses were run in the Rhineland cohort (n=6096) and replication/validation for a subset of lipid traits performed in two independent cohorts (FinnGen, n=7266; EPIC-Potsdam, n=1188). In addition, follow-up analyses were done including PheWAS, two-sample MR (using eQTLs) and gene expression analysis (including mediation). This study has advanced the field by identifying 30 novel genomic loci associated with lipid species, and 25 novel genomic loci associated with fatty acid (FA) composite measures.

Overall, I found this to be a well-presented GWAS the novelty of which comes from the range of lipid traits studied. Compared to established literature, this study uses a more comprehensive approach to lipidomic profiling by evaluated the genetic architecture of the whole plasma FA composition as well as the shared biochemical properties across different lipid species and classes, including their tail length and degree of saturation. However, to maximise the value of this contribution to the literature authors should ensure their reporting meets the relevant community-agreed standards. Related to this, I strongly encourage authors to follow and report against the STREGA checklist for GWAS (<https://www.equator-network.org/reporting-guidelines/strobe-strega/>), the STROBE checklist for observational analyses (<https://www.equator-network.org/reporting-guidelines/strobe/>) and the STROBE-MR checklist for MR (<https://www.equator-network.org/reporting-guidelines/strobe-mr-statement/>). Also related to this, authors should consider use of the LSI Lipidomics Minimal Reporting Checklist (https://lipidomicstandards.org/reporting_checklist/).

Furthermore, analytical code should be made openly available in Github (or similar) and all GWAS summary statistics made openly available.

In general, the methodology appears robust and follows established quality control and methodology standards in GWAS and lipidomic studies. However, some specific queries/points for clarification are listed below.

Major comments:

1. Methods: currently all analyses are adjusted for sex giving sex-averaged results. Given the likely heterogeneity in lipid concentrations and their genetic contributions across males and females, a very useful addition would be to see results of GWAS conducted separately in these subgroups.

Response to point 1:

We thank to reviewer for these valuable comments and agree that sex may modify the association of some genetic variants with lipid species. To address this point we ran a sex-interaction analysis using the fully adjusted model. We only found a significant sex interaction for two lipid species (TAG54:4(FA18:0) and PEP(18:0/20:2)). However, after stratification, we found only one SNP for men to be significant for TAG54:4(FA18:0). We have now included the following in the Results section of the manuscript:

“We additionally tested for evidence of effect modification by sex in the fully adjusted model (model 2). We found strong sex-dependent associations of the rs61763613 and rs11984568 SNPs with PEP(18:0/20:2) ($\beta = -2.54$, $SE = 0.39$, $p \approx 5.18e-11$) and TAG54:4(FA18:0) ($\beta = -0.238$, $SE = 0.035$, $p \approx 1.01e-11$), respectively (Supplementary Figure 4).”

The figures below can clarify this further for the reviewer.

1. TAG54:4(FA18:0)

2. and PEP(18:0/20:2))

2. Results, p.6, heritability estimation:

a. Providing 95% CI for the h^2 estimates would be more informative than SE.

b. Estimates of $h^2=1$ seem inconceivable – isn't it more likely there is an issue with the model convergence and/or assumptions here? Based on Supp Figure 2a – these are outlying values. Could differential LD between associated and not associated regions of the genome be an issue? Also, an $h^2=1$ with a wide 95% CI isn't necessarily meaningful. I suggest these results are checked for robustness as well as greater caution in their reporting.

c. Can you explain the rationale for adjusting for the clinical lipid measurements in the h^2 analysis. It doesn't make sense to me to adjust for heritable traits in this way as then you're adjusting for these at the phenotype level (i.e., accounting for genetic and non-genetic variance). I think this could have unintended consequences on resulting estimates (e.g. bias). For me this makes interpretation of these results challenging

Response to point 2:

- a) We agree with the reviewer and have replaced the previously reported SE values with the corresponding 95% confidence intervals for the heritability estimates.
- b) To address this issue, for those lipid species that exhibited heritability values of 1 (e.g., PE(14:0/18:1)), we also ran LD score regression, which uses summary statistics instead of individual-level data for estimating SNP heritability. However, for these lipids LD score regression yielded inflated heritability estimates of >1 . Although both Genome-wide Complex Trait Analysis (GCTA) and LD score regression take the genomic LD structure into account, it could be that residual population structure may have resulted in inflated heritability estimates for some lipid species. To address this issue, we have added the following as a potential limitation to the discussion section of the manuscript:

"...Although we accounted for genomic LD structure, it could be that residual population structure may have resulted in inflated heritability estimates for some lipid species."

- c) We agree with the reviewer; therefore, all reported heritability estimates are based on lipid species levels adjusted only for age, sex and the first 10 genetic principal components (and thus not for clinical lipid measures).

3. Results, p8: regarding the validation analyses

a. Reporting overall concordance across all three studies would be most informative. Looking at concordance of betas as well as p-values would help to establish whether the relative lack of replication in EPIC-Potsdam is related to power or genuine heterogeneity in signal.

Response to point 3:

- a) We thank the reviewer for this helpful suggestion. Therefore, we have now included scatter plots of SNP effect sizes (β) in the Rhineland study versus those in the replication cohorts (i.e., FinnGen, EPIC Potsdam, Busselton Health Study) in model 1 and model 2 (Extended figure 6 c and d).

- b. Could you also compare results with: <https://www.nature.com/articles/s41467-022-30875-7> (where traits overlap)?

- b) We thank the reviewer for highlighting this important point. Accordingly, we have double-checked all our findings against those reported previously by Cadby et al., Ottensmann et al. and mGWAS-Explorer (<https://www.mgwas.ca/mGWAS/upload/SNPUploadView.xhtml>). Specifically, we compared our results from model 1 with these previously published GWAS summary statistics. This slightly reduced the number of novel loci in our paper to 25. We have marked the known and novel loci using different colors in the tables (**Supplementary Tables 1 to 7**). We also rechecked the tables and ensured that all comparisons are correct. In some cases, there might be an overlap between species, as the study by Cadby et al. only included lipid summary measures (e.g. TAG53:6) but no further details on fatty acid tails that would allow for a precise match with our data (e.g. TAG53:6(FA18:2)). They might also not have the exact same structure, such as our measured CER(24:1) versus

CER(d14:1/24:1)). Similarly, because the association of the *NIPALI* locus with HCER (FA22:1) was not reported previously, we considered this locus to be novel for this lipid species. Thus, to avoid ambiguous comparisons, these types of uncertain matches were not included.

We also checked whether exact lipid species-lead SNP associations at each locus (**Supplementary table 1-4,6,7**) identified in the Rhineland Study replicated in the previous studies (Cadby et al & Ottensmann et al).

Novel mQTLs (**Supplementary table 10-12**), were define as those SNPs that had not previously been associated with any lipid trait or species.

4. Results: the inclusion of some heatmaps or similar to show the phenotypic correlation across traits would be useful.

Response to point 4:

We thank the reviewer for this valuable suggestion. We have now included a heatmap and added the following to the Results section of the revised version of the manuscript under the heading “*Sample characteristics*”:

“*To investigate connections between lipid species, we plotted correlations between lipids in a heatmap (Supplementary Figure 1), most notably showing strong correlations among most TAGs.*”.

5. Results, p9: regarding the MR analysis

a. What correction was made for multiple testing?

Response to point 5a:

We used FDR to correct for multiple testing. This is now clarified as follows:

“Overall, through one-sample MR we identified 43 potentially causal associations between the expression levels of 20 genes and 17 lipid species after adjustment for multiple testing using the FDR method.”

b. There is a need to comment on the potential bias in the causal estimate due to pleiotropy, especially given that some of the MR-Egger intercepts indicate the presence of pleiotropy. In addition, there are many genes where no MR-Egger sensitivity test could be performed and therefore the potential bias due to pleiotropy is unknown. Instrumenting correlated traits like lipids is hard to do – this should be acknowledged.

Response to point 5b:

We ran one-sample MR for 239 candidate genes whose expression levels were available in the Rhineland Study. For these genes, we identified 43 causal associations, of which 22 were also confirmed in two-sample MR. We checked potential pleiotropy effects for all these associations using MR-Egger, which indicated no horizontal pleiotropy for the identified potentially causal associations (**Supplementary table 18**).

6. Results: How was it determined that loci were ‘novel’ (i.e., what criteria were used)?

Response to point 6:

Please see our response to point 3b.

7. Methods, p13, GWAS models: The manuscript could benefit from further clarification regarding the implications of adjusting for clinical lipid measurements in model comparisons.

Response to point 7:

We decided to adjust for clinical lipid measurements as to investigate whether a specific gene was associated with lipid metabolism *in general* or whether it was specific to a certain species. By adjusting for total lipid levels in the blood, results could be interpreted as the balance between lipids (regardless of total lipid concentrations, a specific locus might relate to higher concentrations of one species specifically).

This has been further clarified in the Results section:

“This would also enable us to assess whether the associations between genetic variants and lipid species simply reflect changes in lipid levels in general, or are specific to a certain species.”

And in the Methods section:

“In the second model, we additionally adjusted for fasting status, as well as the levels of total cholesterol, triglycerides, HDL-C, and the use of lipid-lowering medication (model 2). Adjusting for clinical lipid measures enabled us to assess whether genomic loci were associated with overall lipid levels, or were more specifically related to a certain lipid species.”

8. Methods, p15, two-sample MR:

a. How was the adjustment for the traditional lipid measures done? In two-sample MR, the same covars need to be adjusted for in both the SNP-exposure and SNP-outcome GWAS, especially in the case of covars with a heritable component.

Response to point 8a:

Based on a subsequent comment of the Reviewer, we have now included a one-sample MR analysis as the primary method to assess potential causal associations between gene expression and lipid species levels given that it will enable assessment of a larger number of genes. In the one-sample MR analyses we first regressed out the effects of age, sex, levels of total cholesterol, triglycerides, and HDL-C, as well as the use of lipid-lowering medication, fasting status and the first 10 genetic principal components from lipid species levels. The resultant residuals were subsequently normalized using a rank-based inverse normal transformation. This same procedure was performed for gene expression levels by adjusting for the same set of covariates, and in addition, also for batch, red and white blood cell counts, as well as the fraction of basophils, eosinophils, lymphocytes, monocytes, and neutrophils, normalizing the residuals using a rank-based inverse normal transformation. We used lipid species as outcome and gene-expression levels as exposure, where SNP-instruments were defined as those genome-wide significant SNPs that were independently associated with gene-expression levels after clumping. However, in the two-sample MR analyses, where the eQTLs were derived from GTEx, gene expression levels could not be adjusted for clinical lipid measures. Nevertheless, the results of the two-sample MR were highly consistent with those from the one-sample MR, further supporting the validity of our results.

b. This doesn't make sense to me: "Specifically, eQTLs of candidate genes were used as the exposure and variants associated with lipid species levels as the outcome." Instruments (associated SNPs) are selected in the exposure GWAS not the outcome GWAS?

Response to point 8b:

The reviewer is correct that our initial phrasing was unclear: we have indeed selected the genetic instruments from the exposure GWAS. We have now clarified this in the Methods section of the revised manuscript as follows:

“One-sample Mendelian Randomization

We employed a one-sample MR approach to test for evidence that the associations between genes identified through our top-down and bottom-up approach and lipid species (adjusted for clinical lipid measures) were causal, using the OneSampleMR package (version 0.1.5) (64). To this end, gene expression levels were first adjusted for age, sex, batch effects, red and white blood cell counts, and the fractions of basophils, eosinophils, lymphocytes, monocytes, and neutrophils, and residuals were normalized using a rank-based inverse normal transformation. Subsequently, for each gene of interest (exposure), we run a GWAS to identify genome-wide significant SNPs associated with its expression levels that could serve as genetic instruments. Lipid species were treated as the outcome. We clumped SNPs in LD ($r^2 < 0.01$ within a 10 Mb window). Prior to analysis, lipid species levels were adjusted for age, sex, total cholesterol, triglycerides, HDL-C, lipid-lowering medication use, fasting status, and the first 10 genetic PCs. The residuals from this model were also normalized using a rank-based inverse normal transformation. The MR analysis was then performed using two-stage least squares (2SLS) instrumental variable regression. In this framework, the outcome (lipid species) is regressed on covariates and gene expression levels, while gene expression is instrumented by the selected SNPs. This approach allowed us to estimate the causal effect of gene expression on lipid species, while accounting for potential confounding. We utilized the F-statistic as a measure of the strength of SNP-instruments, the Wu-Hausman test p-values for assessing endogeneity, and the Sargan test p-values for checking instrument validity, ensuring that all model assumptions were met. We set the statistical significance level at $FDR < 0.05$ for identifying significant causal associations.

Two-sample Mendelian Randomization

In complementary analyses, we also employed a two-sample MR approach to assess the robustness of the potentially causal associations between the genes and lipid species identified in our one-sample MR analyses, using the TwoSample MR package; because compared to one-sample MR, two-sample MR is less prone to overfitting. We downloaded whole blood eQTL data from the GTEx resource (57), and used variants associated with gene expression levels (i.e., eQTLs) as genetic instruments in the MR analyses. We clumped SNPs in LD ($r^2 < 0.01$ within a 10 Mb window). The method selection for Two-sample MR was based on the number of SNP instruments available, including Wald Ratio (1 SNP instrument), inverse variance weighted method (≥ 2 SNP instruments), and MR-Egger method (> 3 SNP instruments). To assess the robustness of our MR results, we performed several sensitivity analyses, such as calculating F-statistics to assess the risk of weak-instrumental bias and MR-Egger intercept to identify horizontal pleiotropy.”

c. Given you have expression data in the cohort, could you do a one-sample MR?

Response to point 8c:

We thank the reviewer for this valuable suggestion. Following the reviewer's suggestion, we have now also performed one-sample MR analyses as described above under our responses to Points 8a & 8b.

9. Methods, p15, mediation analysis: please could you provide more detail of the approach and model used for the mediation analysis. It's not obvious how the 'lavaan' package would have been used for mediation? Nor do I

understand what the mediation ‘estimate’ in Table 2 is or how it relates to mediation results in Suppl Data 14. Providing the code (as mentioned above) could help clarify.

Response to point 9:

We thank the reviewer for this helpful comment. We have now expanded the section about mediation analysis to include more details on the model and approach used with the lavaan package. Specifically, we have added the following elaboration under the heading “*Functional validation through gene expression analysis*” of the Methods section:

*“... Lastly, for combinations where the gene-lipid and SNP-gene associations were significant, we conducted mediation analyses using the R package lavaan (v.0.6-11) to assess whether gene expression levels (mediator) mediated the relationship between the lead SNPs (exposure) and lipid species levels (outcome). Specifically, we performed structural equation modeling (SEM), using a standard three-variable mediation model: path a represented the effect of the SNP on the gene expression, path b the effect of the gene expression on lipid levels, and path c’ the direct effect of the SNP on lipid levels. Indirect effects were calculated as a*b, the direct effects as c’, and the total effects as a*b + c’. The model was specified in lavaan using the sem.fit function, using bootstrapping (with 1,000 resamples) for calculating confidence intervals. For the mediation analyses, we used the residuals of gene expression adjusted for age, sex, batch, and blood cell counts, and residuals of lipid species adjusted for age, sex, the first 10 principal components, total cholesterol, triglycerides, HDL-C, lipid-lowering medication use, and fasting status.”.*

10. Discussion, p11: here it says, “Results of our PheWAS further highlight the important role of complex lipids as modifiers of a large number of different traits and diseases.” However, there has been no mention of the potential bias due to reverse causality here, that the disease phenotype may be driving changes in lipid metabolism and therefore the lipid profile and FA composition. This need addressing in the discussion.

Response to point 10:

We thank the reviewer for highlighting this important point. We agree with that diseases might also lead to changes in lipid metabolism. However, the analyses were performed in a relatively healthy general population-based sample largely free from chronic diseases. Moreover, the results of our reverse MR analyses did not yield convincing evidence for this latter possibility. As such, reverse causality seems to be a less likely explanation for our findings. We have now addressed this issue in the discussion section of the revised manuscript by adding the following elaboration:

“... It has to be noted though that certain diseases could also lead to changes in lipid profiles, rather than vice versa. However, we believe this to be less likely as the associations were identified in a mostly healthy population. Moreover, the results of our reverse MR analyses did not yield convincing evidence for this latter possibility...”.

Minor comments:

11. Results, p4: I don’t completely understand the approach to defining loci and lead SNPs as described here (“These genomic loci were identified ... significant SNPS”) and in the corresponding section of the Methods (p13). Can you reword and/or add a diagram (in suppl) to help?

We agree with the reviewer that our initial phrasing was confusing. Therefore, we have reworded this section as follows to make it clearer:

“... These genomic loci were identified by merging all metabolome-wide significant SNPs across all lipid species ($p < 2.27E-10$), followed by linkage disequilibrium (LD) clumping to identify independent significant SNPs (using $r^2 \geq 0.6$ for defining independent significant SNPs) within 250 kb of each other, and 218 lead SNPs (using $r^2 \geq 0.1$ for the clumping of independent significant SNPs).”

12. Results, p.4: define what the % in the brackets represents in the sentence: “The lipid species within the sphingomyelins (SMs) (100%)”

We have now clarified this by including the denominator:

“The lipid species within the sphingomyelins (SMs) (100% of 12 species), TAGs (78.2% of 518 species) and DAGs (75.9% of 58 species) classes had the highest number of genetic associations.”

We have also clarified this in a similar sentence in the results:

“Across all different lengths, lipids carrying a FA tail of 24 (81.2% of 16 24-carbon carrying composites), 20 (74.2% of 65 20-carbon carrying composites) or 22 (63.6% of 54 22-carbon carrying composites) carbons had the highest percentage of significant genetic associations (**Figure 2**).”

13. Several of the Supplementary Data Tables (in the excel file) need additional annotation to help interpretation. For example, in sp1, it’s not clear to me what the three results ‘chunks’ represent, save for the fact one of them seems to be from FinnGen. Same for sp3 & sp4.

We have adjusted the Supplementary data files as suggested.

14. Results, figure 3: I don’t find any elements of this figure particularly informative – in particular (b). For a, c, d, would boxplots showing the distribution of correlation coefficient by class be more effective?! Same applies to suppl figure 1.

Based on the reviewer’s suggestions, we have modified Figure 3. The previous subplots (i.e., Figure 3 a, c and d) have been plotted into a single figure with correction of the axis for clarity, and the previous subplot 3b has been replaced with forest plots depicting the genetic correlations for each class across different lipid categories.

Genetic Correlation: HDL-C

Genetic Correlation: HDL-C

15. Results, supp data 5: shading at the legend at the top is suggested but seemingly not applied. Columns J & K seem to be indicating which model the SNP/lipid association comes from but what about where they appear in both models?

We have critically assessed all figures and adjusted them where applicable. We have applied the shading to indicate the beta and p-values of the required models. We also included the beta and p-values for SNP-lipid species associations that are significant in both model 1 and 2 (**Supplementary table 5**).

16. Results, Supp Figure 2a: this would be easier to follow if the lipid classes were presented in the same order in the two plots or if the two plots were combined into a two-way boxplot where boxes for model 1 & 2 for the same lipid were next to each other.

We have critically assessed all figures and adjusted them where applicable.

17. Results, p7: in the top paragraph where it says: “After additional adjustment for clinical lipid measurements ...” – I don’t understand what analysis this (and the rest of this paragraph) is referring to because I thought the whole paragraph was about combined model 1 and 2 results.

This whole paragraph indeed refers to the results of the model adjusted for clinical lipid measures (model 2) as mentioned in the opening sentence of the paragraph. Therefore, the subordinate clause “After additional adjustment for clinical lipid measures...” in the middle of this paragraph is indeed redundant and has now been removed from the revised version of the manuscript.

18. Results, p7: re the GTEx analysis – please specify tissue type.

We used all the available 48 tissues from GTEX v8 database. We have included this information in the revised version of the manuscript as follows:

“and we used all the available 48 tissues from GTEX v8 database”

19. Results, p.7, second paragraph: I’m not what analysis the final sentence of this paragraph is related to (“Additionally, we identified 64 and 82 mQTLs ...”) – it doesn’t seem to fit with the rest of the paragraph which is about GTEx data and eQTLs.

This last part indeed refers to mQTLs instead of eQTLs. To clarify this distinction, we have now included this part in a separate subsequent paragraph as follows:

“In addition to the eQTLs, we also identified 65 and 80 mQTLs which were associated with the lipid species in model 1 and 2, respectively, which were also detected in previous lipidomic GWAS studies (26).”

20. Results, supp data 12 – what does the highlighting indicate? Add key/legend.

In the modified excel file, supplementary data 15, the highlight indicates the overlap of lipid species available within each cohort. We indicated this now in the excel file.

21. Results, p8: check reference to supp data 1 at the top of this page – is this correct?

We thank the reviewer for pointing this out. We indeed refer to supplementary data 1 to show the replication of model 1 results with lipid species and check the replication with EPIC Potsdam cohort indicated in supplementary data 1.

22. Results, suppl data 10: could you add the associated lipid traits to this table?

We thank the reviewer for this comment. In the modified excel file, the supplementary data has been moved to supplementary data 13. We could add the lipid traits, however the input for PheWAS lookups was based only on the lead SNPs from model 2.

23. Results, suppl data 14: further explanation of the results here is needed. Is this all based on the observational analysis of expression data? And should there be a gap between the tables in columns A-H and I-Q or is this a continuation in which case what's the relation between the genes in the first part of the table (A-H) and the genes/SNPs in the second part (I-Q)?

We thank the reviewer for pointing this out. This data has been moved to supplementary data 17 in the modified supplementary data. This is indeed based on the observational analysis of “Associations of primary candidate genes with lead SNPs” and followed by “Associations of primary candidate genes with lipid species”. Mediation analysis is conducted based on the overlap of significant associations between associations of primary candidate genes with lead SNPs and associations of primary candidate genes with lipid species. Row-wise there aren't relations among “A-J”, “K-S” and “T-AD”. For each section, we sort the association results based on decreasing p-values.

24. Results, p9: not sure how ‘top modules’ have been defined – top based on what?

We thank the reviewer for pointing this out. Due to addition of gene-expression analysis, one sample and Two-sample MR associations with gene-expression, we removed the WGNA analysis throughout the manuscript.

25. Method, p13: what is the rationale behind performing a transformation on the data (residuals) after having fitted a linear model (that assumes a normal distribution) – if the raw lipid data were not normally distributed, shouldn't they be transformed prior to regressing out the fixed effects?

This approach was chosen to enable better comparability of our findings with those from previous complex lipid GWAS studies. Specifically, in the FinnGen study by Ottensmann et al., which we used for replication of our findings, a similar statistical approach was applied, which the authors describe as follows:

Adjusting for clinical lipid measures enabled us to assess whether genomic loci were associated with overall lipid levels, or were more specifically related to a certain lipid species. Residuals obtained in either model were subsequently normalized using a rank-based inverse normal transformation before performing the GWAS.”

26. Method, p.13: Can the authors comment on use of the somewhat lenient info score threshold of 0.3 – why was this cut off chosen and what are the implications?

Here too, the choice for this threshold for the imputation score was based on enabling better comparability of our findings with those of previous complex lipid GWAS studies. Specifically, in the GWAS study by Cadby et al. the same imputation score quality threshold of 0.3 was applied.

27. Discussion, p12: worth mentioning that a limitation of the validation analysis is that many associations could not be tested in either cohort.

Indeed, one of the pitfalls of an extensive lipid panel is that there might be more unique species that are not measurable in most people. As such, some associations could not be tested in the other replication cohort that were either smaller or used a less comprehensive lipidomics platform.

We now included this in the discussion:

“... Despite not being able to do this for all associations on account of missing data in the other cohorts, which were either smaller or used a less comprehensive lipidomics platform, we could validate most of our key findings.”

28. Discussion: Possible impact of variations in lipid measurement techniques across cohorts on validation/concordance with literature should be discussed.

We have now included a brief discussion of this issue in the limitations section of the manuscript:

“The extensive unique complex lipids data on a large number of individuals is a main strength of our study, as we could investigate the genetics of specific species, as well as the whole FA composition, which had been challenging so far. In particular, this was further strengthened by combining data from lipids measured on the same platform in both the Rhineland Study and EPIC-Potsdam cohort. However, some of our findings could not be directly compared to those of previous GWAS studies, because of differences in lipidomics platforms used in each study.”

29. Discussion: Clinical Relevance: A deeper discussion on how the identified loci might inform clinical interventions or risk stratification would strengthen the manuscript's translational relevance.

We have now included examples of the clinical relevance of our results in the discussion:

“These results could pave the way to precision medicine and improved risk stratification. For example, measuring the identified lipids could not only aid in identifying individuals with a risk allele – without complex genetic analyses –, but could possibly predict disease better than the traditional lipid measurements in disease risk models. Furthermore, by identifying individuals with a genetic risk factor, stricter monitoring and regulating of these identified lipids could possibly aid in preventing disease. Therewith, it could improve precision medicine. In addition to the possible gene-lipid-disease links,...”

Reviewer #3 (Remarks to the Author):

REVIEWER COMMENTS

Reviewer #1 (Remarks to the Author):

I appreciate the extensive work that the authors have put into this revision.

Comment 1:

However, based on the additional data provided, I still have major concerns and must get back to the third point I raised in my previous review. There, I particularly questioned the reliability of loci defined by a single SNP association, which are furthermore not replicated in any of the other studies investigated by the authors.

While reviewing the regional association plots provided in this revision, I was surprised by the large number of single SNP associations reported. If there is only one SNP found to be associated, but the other SNPs in the locus are not showing any association (or weaker/insignificant association but no LD), it is highly likely that this is a false positive. This is even more likely in cases where one SNP shows an association and other SNPs in LD do not even show a trend towards association. This becomes an issue, as I counted around 20 such cases in both the model 1 and model 2 discovery studies, which is a third of all associations. Even more concerning is that many of these are loci reported to be novel.

This observation casts doubt on the correctness of the reported results, which led me to investigate all parts and data provided in the revised version in detail. This is time-consuming, and will take more time. However, this issue - for me - must be resolved, and can be resolved while I continue to review other aspects of this work. I should note in this regard that I personally see the data used and the study by itself as valuable and impactful, and that this assessment will not change because fewer loci are reported. But the reported findings should be correct to the best knowledge of all people involved.

I hence urge the authors to either revisit their methodological approach or at least provide the following information in addition for all associations:

a) the number of significant SNPs per association in a locus (you now provided the number of SNPs in LD, which by itself is not sufficient).

Response to comment 1:

We thank the reviewer for these insightful comments and fully agree that it is important to explicitly highlight the number of genetic variants on which each reported association is based. Therefore, we have now included two new columns, entitled “nSNPs” and “nGWASSNPs”, to Supplementary Tables 1–7. For each independent SNP these columns provide the total number of SNPs in linkage disequilibrium (LD) and the number of genome-wide significant ($P < 5 \times 10^{-8}$) SNPs within the corresponding locus, respectively. We acknowledge that certain loci (e.g., loci 4, 9, 25, 26, 27, 28, 34, 38, 39, and 42 in Supplementary data 1) contain relatively few SNPs that are either in LD or reach genome-wide significance. Notably, loci 27 and 28 each contain only a single SNP in LD within the genomic region, but are likely true associations as they have been identified in earlier lipidomics GWAS studies as well. Specifically, locus 27 (lead SNP rs9411262) was previously reported to be associated with phosphatidylcholine (16:0/22:5n3, 18:1/20:4) levels in the GWAS “Rare and common genetic determinants of metabolic individuality and their effects on human health” [10.1038/s41591-022-02046-0]. Similarly, locus 28 (SNP rs603424) was previously associated with lysophosphatidylcholine acyl C16:1 levels in the GWAS “A cross-platform approach identifies genetic regulators of human metabolism and health” [10.1038/s41588-020-00751-5].

Comment 2:

b) the information if significant SNPs were imputed or genotyped directly. If imputed, the info score should also be provided.

We agree with the reviewer that it is important to include the information on whether the significant SNPs were genotyped or imputed. Therefore, as suggested by the reviewer, we have added two new columns, entitled “Rsq” and “Genotyped”, to Supplementary Data 1–4. For each independent SNP these columns provide the imputation quality score (Rsq) and an indication of whether the SNP was directly genotyped or imputed (Genotyped).

Comment 3:

I was further asked by the editor to particularly review the methods for generating and processing the lipidomics

data. There are two aspects that raise concerns:

a) Lipidomics data is often log-normally distributed, which requires log-transformation even if data is quantitative. The manuscript does not state transformation of lipid concentrations. I would advise to investigate if the distributions of lipid concentrations are skewed or normal. If they are (close to) normally distributed, this should be stated. Otherwise, I would advise to perform log-transformation.

Response to comment 3:

We thank the reviewer for this valuable comment. The reviewer is indeed correct to point out that the concentrations of many of these lipids are not normally distributed (please find below some distribution plots for illustration). To account for these skewed distributions all our GWAS analyses were already performed after a rank-based inverse normal transformation of the residuals (after regressing out the effects of age, sex and the first 10 PCs in model 1, and additionally for total cholesterol, triglycerides, HDL-C and the use of lipid-lowering medication in model 2), as described in the methods section (under GWAS). This method enforces a normal distribution and was chosen to ensure consistency with previous GWAS studies on lipid levels (Cadby G, Giles C, Melton PE, Huynh K, Mellett NA, Duong T, et al. Comprehensive genetic analysis of the human lipidome identifies loci associated with lipid homeostasis with links to coronary artery disease. Nat Commun. 2022 Jun 6;13(1):3124; McCaw ZR, Lane JM, Saxena R, Redline S, Lin X. Operating characteristics of the rank-based inverse normal transformation for quantitative trait analysis in genome-wide association studies. Biometrics. 2020 Dec;76(4):1262-1272. doi: 10.1111/biom.13214; Barber MJ, Mangravite LM, Hyde CL, Chasman DI, Smith JD, McCarty CA, Li X, Wilke RA, Rieder MJ, Williams PT, Ridker PM, Chatterjee A, Rotter JI, Nickerson DA, Stephens M, Krauss RM. Genome-wide association of lipid-lowering response to statins in combined study populations. PLoS One. 2010 Mar 22;5(3):e9763. doi: 10.1371/journal.pone).

Comment 4:

b) Inclusion of lipids present in at least 1% of the population is a very relaxed threshold. Based on the total number of participants, this includes GWASs with around 60 individuals. Aside from low power, a high amount of missing data in lipid measurements can point to issues in measurement reliability. I would ask the authors to revisit this approach, or at least provide some rationale on why they used this threshold.

I apologize for missing this in my first review.

Response to comment 4:

The reviewer is correct to point out that missingness in lipidomic data is an important issue that should be considered carefully. Therefore, to clarify the degree of missingness for each of the lipid species included in our study, we have now included an additional table summarizing the frequency of missingness for each lipid species (Supplementary Table 7). For example, as shown in the table, there are only 37 lipids that would be excluded based on a more stringent threshold of 10%. However, this is unlikely to be due to issues with measurement reliability: the lipids with a high degree of missingness have previously been reported to only have very low concentrations in human plasma. In a general population, we would therefore expect the concentrations of these specific lipid species to be below the limit of detection in a substantial number of individuals.

Reviewer #2 (Remarks to the Author):

This manuscript has improved substantially following review and I thank the authors for their comprehensive response to my previous comments. Below are some outstanding points for consideration.

Comment 1:

Major:

1. As results of your GWAS suggest, many mQTL are non-specific in that they show associations across a range of lipid traits. As such, in many cases (where you perform 2-sample MR with lipid traits as exposures) it is unrealistic to expect to be able to instrument single lipid traits in isolation. That is to say, even where you have a seemingly robust set of instruments for lipid trait ‘a’, those same SNPs may well proxy other lipid traits (both measured and unmeasured) in the dataset (i.e. they are shared instruments). I think this point is at the very least worth acknowledging in the discussion.

Response to comment 1:

We thank the reviewer for highlighting this important issue. We agree that some SNP instruments were shared across multiple lipid traits, making it challenging to unequivocally delineate the potentially causal associations of individual lipid species with other traits in 2-sample MR. We have now acknowledged this issue as a limitation in the discussion by stating:

"Although our analyses identified several potentially causal associations between certain lipid species and specific traits, some SNP instruments were shared across multiple lipid species. For lipid species that were instrumented by sets of overlapping SNPs, it is therefore challenging to determine true causal relationships between individual lipid species and specific traits. Future larger studies using multivariable MR methods may help account for the genetic correlation between different lipid traits (45,46)."

Comment 2:

Minor:

2. I couldn't see that the abbreviations in Suppl. Fig 1 were defined anywhere.

Response to comment 2:

We thank the reviewer for noticing this. We had indicated the abbreviations of lipid classes for the complex lipids in Supplementary Table 3; however, for clarity and convenience, we have now also included them in the legend of Supplementary Figure 1.

Comment 3:

3. I couldn't see a reference to Supplementary data 9 & 13 (apologies if I missed it).

Response to comment 3:

Indeed, there was a typo in the manuscript. We have now corrected this in the respective sections.

"After adjustment for clinical lipid measurements, use of lipid-lowering medication and fasting status, the effect estimates for the lipid species levels changed in a class-dependent manner (Figure 3a, Supplementary Data 9)"

"Thus, these genes might affect disease phenotypes through changes in the lipid metabolism. Additional information can be found in Supplementary Data 13"

Comment 4:

4. "In addition to the eQTLs, we also identified 65 and 80 mQTLs which were associated with the lipid species in model 1 and 2, respectively, which were also detected in previous lipidomic GWAS studies (26)." I still think this sentence is in the wrong place. And I don't understand why you are singling out the '65 and 80 mQTLs' when the whole analysis is about identifying mQTLs.

Response to comment 4:

We thank the reviewer for this valuable comment. To clarify, metabolite quantitative trait loci (mQTLs) represent genetic variants that are significantly associated with metabolite or lipid levels. Identifying mQTLs is indeed a key objective of our GWAS because these loci provide insights into the genetic regulation of lipid metabolism and help bridge the gap between genetic variation and metabolic phenotypes.

In our study, we highlighted the 65 and 80 mQTLs to specifically emphasize the subset of loci that were not only genome-wide significant in our analysis but also replicated or were previously reported in independent lipidomic GWAS studies. We believe that this demonstrates the reproducibility and biological relevance of our findings across two independent models. We have revised the text in the Results section for clarity in the current version:

"...For further validation of our findings, in addition to the eQTLs, we also assessed which of the lead SNPs associated with lipid species levels – which thus could be classified as metabolite quantitative trait loci (mQTLs) – were also reported previously. Sixty-five out of the 218 and 80 out of the 217 mQTLs identified in our analyses in model 1 and 2, respectively, were also detected in previous lipidomic GWAS studies (26)."

Comment 5:

5. L257, should this be referencing Suppl Fig 6a (not 3a)?

Response to comment 5:

We thank the reviewer for pointing this out. There was indeed a typo in mentioning the supplementary figure in the manuscript. We have now corrected this as follows:

"Of the corresponding associations, 175 were replicated in the Rhineland Study, while only 33 were not (Supplementary Figure 6a, Supplementary Data 14)."

Comment 6:

6. L302 – check this reference to Suppl Data 15 – not sure it is correct.

Response to comment 6:

We thank the reviewer for pointing this out. We have now corrected this as follows:

*“Thus, these genes might affect disease phenotypes through changes in the lipid metabolism. Additional information can be found in **Supplementary Data 13**”*

Comment 7:

7. Still some of the suppl data tables could do with additional column headers and/or labels – eg Supp data 16 – different betas need explanations, supp data 18 also.

Response to comment 7:

We thank the reviewer for pointing out this important issue. We have now added footnotes explaining the columns in each table within the respective supplementary datasets.

Comment 8:

8. L538-540: repeated sentence.

Response to comment 8:

We thank the reviewer for pointing this out. We have therefore removed the repeated sentence

“RNA integrity and quantity were evaluated using the tapestation RNA assay on a Tapestation4200 instrument (Agilent)”

Comment 9:

9. For clarity, in the GWAS, did you analyse dosages/probabilities/bestguess genotypes?

Response to comment 9:

In our GWAS analyses, we used imputed allele dosages as genotype input. Specifically, we used the rvtsts software with the --dosage DS flag, where “DS” represents the imputed allele dosage values derived from the imputed SNP VCF file. We have included this information to the Methods section under the heading ‘GWAS’ as follows:

“We performed a GWAS on the absolute concentrations of lipid species and FA composite measures with Rvtsts using imputed allele dosages as genotype input (51).”